# NODE EMBEDDING FROM NEURAL HAMILTONIAN ORBITS IN GRAPH NEURAL NETWORKS

## ABSTRACT

In the graph node embedding problem, embedding spaces can vary significantly for different data types, leading to the need for different GNN model types. In this paper, we model the embedding update of a node feature as a Hamiltonian orbit over time. Since the Hamiltonian orbits generalize the hyperbolic exponential maps, this approach allows us to learn the underlying manifold of the graph in training, in contrast to most of the existing literature that assumes a fixed graph embedding manifold. Our proposed node embedding strategy can automatically learn, without extensive tuning, the underlying geometry of any given graph dataset even if it has diverse geometries. We test Hamiltonian functions of different forms and verify the performance of our approach on two graph node embedding downstream tasks: node classification and link prediction. Numerical experiments demonstrate that our approach adapts better to different types of graph datasets than popular state-of-the-art graph node embedding GNNs.

## 1 INTRODUCTION

Graph neural networks (GNNs) (Yue et al., 2019; Ashoor et al., 2020; Kipf & Welling, 2017b; Zhang et al., 2022; Wu et al., 2021) have achieved good inference performance on graph-structured data such as social media networks, citation networks, and molecular graphs in chemistry. Most existing GNNs embed graph nodes in Euclidean spaces without further consideration of the dataset graph geometry. For some graph structures like the tree-like graphs (Liu et al., 2019), the Euclidean space may not be a proper choice for the node embedding. Recently, hyperbolic GNNs (Chami et al., 2019; Liu et al., 2019) propose to embed nodes into a hyperbolic space instead of the conventional Euclidean space. It has been shown that tree-like graphs can be inferred more accurately by hyperbolic GNNs. Furthermore, works like Zhu et al. (2020b) have attempted to embed graph nodes in a mixture of the Euclidean and hyperbolic spaces, where the intrinsic graph local geometry is attained from the mixing weight.

Embedding nodes in a hyperbolic space is achieved through the exponential map (Chami et al., 2019), which is essentially a geodesic curve on the hyperbolic manifold as the projected curve of the cogeodesic orbits on the manifold's cotangent bundle (Lee, 2013; Klingenberg, 2011). In our work, we propose to embed the nodes, via more general Hamiltonian orbits, into a general manifold, which generalizes the hyperbolic embedding space, i.e., a strongly constrained Riemannian manifold of constant sectional curvature equal to $-1$.

From the physics perspective, the cotangent bundles are the natural phase spaces in classical mechanics (De León & Rodrigues, 2011) where the physical system evolves according to the basic laws of physics modeled as differential equations on the phase spaces. In this paper, we propose a new GNN paradigm based on Hamiltonian mechanics (Goldstein et al., 2001) with flexible Hamiltonian functions. Our objective is to design a new node embedding strategy that can automatically learn, without extensive tuning, the underlying geometry of any given graph dataset even if it has diverse geometries. We enable the node features to evolve on the manifold under the influence of neighbors. The learnable Hamiltonian function on the manifold guides the node embedding evolution to follow a learnable law analogous to basic physical laws.

**Main contributions.** Our main contributions are summarized as follows:
1. We take the graph as a discretization of an underlying manifold and enable node embedding through a learnable Hamiltonian orbit associated with the Hamiltonian scalar function on its cotangent bundle.

2. Our node embedding strategy can automatically learn, without extensive tuning, the underlying geometry of any given graph dataset even if it has diverse geometries. We empirically demonstrate its ability by testing on two graph node embedding downstream tasks: node classification and link prediction.

3. From empirical experiments, we observe that the oversmoothing problem of GNNs can be mitigated if the node features evolve through Hamiltonian orbits. By the conservative nature of the Hamiltonian equations, our model enables a stable training and inference process while updating the node features over time and layers.

## 2   RELATED WORK

While our paper is related to Hamiltonian neural networks in the literature, we are the first, to our best knowledge, to model graph-structured data with Hamiltonian equations. In what follows, we briefly review Hamiltonian neural networks, Riemannian manifold GNNs, and physics-inspired GNNs.

**Hamiltonian neural networks.** Among these physics-inspired deep learning approaches, Hamiltonian equations have been applied to conserve an energy-like quantity when training neural networks. The papers Greydanus et al. (2019); Zhong et al. (2020); Chen et al. (2021) train a neural network to infer the Hamiltonian dynamics of a physical system, where the Hamiltonian equations are solved using neural ODE solvers. The work Haber & Ruthotto (2017) builds a Hamiltonian-inspired neural ODE to stabilize the gradients so as to avoid vanishing and exploding gradients. The paper Huang et al. (2022) further studies the adversarial robustness of Hamiltonian ODE. *In this paper, we focus on applying Hamiltonian equations to graph neural networks, which has not been investigated in the above-mentioned works.*

**Riemannian manifold GNNs.** Most GNNs in the literature Yue et al. (2019); Ashoor et al. (2020); Kipf & Welling (2017b); Zhang et al. (2022); Wu et al. (2021) embed graph nodes in Euclidean spaces. In what follows, we simply call them (vanilla) GNNs. They perform well on some datasets like the Cora dataset McCallum et al. (2004) whose $\delta$-hyperbolicity Chami et al. (2019) is high. When dealing with the datasets whose $\delta$-hyperbolicity are low (hence embedding should more appropriately be in a hyperbolic space) such as the Disease Chami et al. (2019) and Airport Chami et al. (2019) datasets, those GNNs suffer from improper node embedding. To better handle hierarchical graph data, (Liu et al., 2019; Chami et al., 2019; Zhang et al., 2021b; Zhu et al., 2020b) propose to embed nodes into a hyperbolic space, thus yielding hyperbolic GNNs. Moreover, Zhu et al. (2020b) proposes a mixture of embeddings from Euclidean and hyperbolic spaces. This mixing operation relaxes the strong space assumption of using only one type of space for a dataset. *In this paper, we embed nodes into a general learnable manifold via the Hamiltonian orbit on its symplectic cotangent bundle.* This allows our model to flexibly adapt to the inherent geometry of the dataset.

**Graph Neural diffusion:** Neural Partial Differential Equations (PDEs) have been applied to graph-structured data (Chamberlain et al., 2021b;a; Song et al., 2022), where different diffusion schemes are assumed when performing message passing on graphs. To be more specific, the heat diffusion model is assumed in (Chamberlain et al., 2021b) and the Beltrami diffusion model is assumed in (Chamberlain et al., 2021a; Song et al., 2022). (Rusch et al., 2022) models the nodes in the graph as coupled oscillators, i.e., a second-order ODE. *While above mentioned graph neural diffusion schemes and our model all use ODEs, there is a fundamental difference between our model and graph neural flows. In the graph PDEs, they wrap the message passing function, e.g., constant aggregation function like the one in GCN, and attention-based aggregation function like the one in GAT, into an ODE function. In contrast, our model treats the node embedding process and node aggregation process as two independent processes, where we use the ODE function only to learn a suitable node embedding space which is then followed by a node aggregation step. To sum up, our ODE is actually a node embedding layer taking node features as the input whereas graph PDEs are node aggregation layers taking node features as well as graph adjacency matrix as the input.*
*Notations:* We use the *Einstein summation convention* (Lee, 2013) for expressions with tensor indices.

## 3   MOTIVATIONS AND PRELIMINARIES

In this section, we briefly review the concepts of the geodesic curve on a Riemannian manifold from the principle of stationary action in the form of Lagrange's equations. We then further generalize the geodesic curve to the Hamiltonian orbit associated with an energy function $H$, which is a conserved quantity along the orbit. We first summarize the motivation of our work as follows.

**Motivation I: from the hyperbolic exponential map to Riemannian geodesic.** The geodesic curve gives rise to the exponential map that maps points from the tangent space to the manifold and has been

utilized in (Chami et al., 2019) to enable graph node embedding in a special Riemannian manifold known as the hyperbolic space. From this perspective, by using the geodesic curve, we generalize the graph node embedding to an arbitrary (pseudo-)Riemannian manifold with **learnable local geometry** $g$ using Lagrange's equations.

**Motivation II: from Riemannian geodesic to Hamiltonian orbit.** Despite the above conceptual generalization for node embedding using geodesic curves, the specific curve formulation involving minimization of curve length may result in a loss of generality for node feature evolution along the curve. We thus further generalize the geodesic curve to the Hamiltonian orbit associated with an energy function $H$ that is conserved along the orbit. In Section 4, we propose graph node embedding without an explicit metric by using Hamiltonian orbits with **learnable energy functions** $H$.

### 3.1 MANIFOLD AND RIEMANNIAN METRIC

**Manifold and local chart representation.** On a $d$-dimensional manifold $M$, for each point on $M$, there exists a triple $\{q, U, V\}$, called a chart, such that $U$ is an open neighborhood of the point in $M$, $V$ is an open subset of $\mathbb{R}^d$, and $q : U \to V$ is a homeomorphism, which gives us a coordinate representation for a local area in $M$.

**Tangent and cotangent vector spaces.** For any point $q$ on $M$ (we identify each point covered by a local chart on $M$ by its representation $q$), we may assign two vector spaces named the tangent vector space $T_q M$ and cotangent vector space $T_q^* M$. The vectors from the tangent and cotangent spaces can be interpreted as representing a velocity and generalized momentum of motion in classical mechanics, respectively.

**Riemannian metric.** A Riemannian manifold is a manifold $M$ equipped with a *Riemannian metric* $g$, where we assign to any point $q \in M$ and pair of vectors $u, v \in T_q M$ an *inner product* $\langle u, v \rangle_{g(q)}$. This assignment is assumed to be smooth with respect to the base point $q \in M$. The length of a tangent vector $u \in T_q M$ is then defined as

$$\|u\|_{g(q)} := \langle u, u \rangle_{g(q)}^{1/2}. \tag{1}$$

- **Local coordinates representation:** In local coordinates with $q = (q^1, \ldots, q^d)^\mathsf{T} \in M, u = (u^1, \ldots, u^d)^\mathsf{T} \in T_q M$ and $v = (v^1, \ldots, v^d)^\mathsf{T} \in T_q M$, the Riemannian metric $g = g(q)$ is a real symmetric *positive definite* matrix and the inner product above is given by

$$\langle u, v \rangle_{g(q)} := g_{ij}(q) u^i v^j \tag{2}$$

- **Pseudo-Riemannian metric:** We may generalize the Riemannian metric to a metric tensor that only requires a *non-degenerate* condition (Lee, 2018) instead of the stringent positive definiteness condition in the inner product. One example of a pseudo-Riemannian manifold is the Lorentzian manifold, which is important in applications of general relativity.

### 3.2 GEODESIC CURVES AND EXPONENTIAL MAPS

**Length and energy of a curve.** Let $q : [a, b] \to M$ be a smooth curve.[1] We define the following:

- length of the curve: $$\ell(q) := \int_a^b \|\dot{q}(t)\|_{g(q(t))} \, \mathrm{d}t. \tag{3}$$

- energy of the curve: $$E(q) := \frac{1}{2} \int_a^b \|\dot{q}(t)\|_{g(q(t))}^2 \, \mathrm{d}t. \tag{4}$$

**Geodesic curves.** On a Riemannian manifold, geodesic curves are defined as curves that have a minimal length as given by (3) and with two fixed endpoints $q(a)$ and $q(b)$. However, computations based on minimizing the length to obtain the curves are difficult. It turns out that the minimizers of $E(q)$ also minimize $\ell(q)$ (Malham, 2016). Consequently, the geodesic curve formulation may be obtained by minimizing the energy of a smooth curve on $M$.

**Principle of stationary action and Euler–Lagrange equation.** The Lagrangian function $L(q(t), \dot{q}(t))$ minimizes the following functional (in physics, the functional is known as an **action**)

$$S(q) = \int_a^b L(q(t), \dot{q}(t)) \mathrm{d}t. \tag{5}$$

---

[1]We abuse notations in denoting the chart coordinate map as $q$ and the curve as $q(t)$. It will be clear from the context which one is being referred to.

with two fixed endpoints at $t = a$ and $t = b$ only if the following *Euler–Lagrange equation* is satisfied:

$$\frac{\partial L}{\partial q^i}(q(t), \dot{q}(t)) - \frac{\mathrm{d}}{\mathrm{d}t}\frac{\partial L}{\partial \dot{q}^i}(q(t), \dot{q}(t)) = 0. \tag{6}$$

**Geodesic equation for geodesic curves.** The Euler–Lagrange equation derived from minimizing the energy (4) with local coordinates representation,

$$L = \frac{1}{2}\|\dot{q}(t)\|^2_{g(q(t))} = \frac{1}{2}g_{ik}(q)\dot{q}^i\dot{q}^k \tag{7}$$

is expressed as the following ordinary differential equations called the *geodesic equation*:

$$\ddot{q}^i + \Gamma^i_{jk}\dot{q}^j\dot{q}^k = 0, \tag{8}$$

for all $i = 1, \ldots, d$, where the Christoffel symbols $\Gamma^i_{jk} = \frac{1}{2}g^{i\ell}\left(\frac{\partial g_{\ell j}}{\partial q_k} + \frac{\partial g_{k\ell}}{\partial q_j} - \frac{\partial g_{jk}}{\partial q_\ell}\right)$ and $[g^{ij}]$ denotes the inverse matrix of the matrix $[g_{ij}]$. The solutions to the geodesic equation (8) give us the geodesic curves.

**Exponential map.** Given the geodesic curves, at each point $x \in M$, for velocity vector $v \in T_x\mathcal{M}$, the *exponential map* is defined to obtain the point on $M$ reached by the unique geodesic that passes through $x$ with velocity $v$ at time $t = 1$ (Lee, 2018). Formally, we have

$$\exp_x(v) = \gamma(1) \tag{9}$$

where $\gamma(t)$ is the curve given by the geodesic equation (8) with initial conditions $q(0) = x$ and $\dot{q}(0) = v$.

With regards to **Motivation I**, we note that (Chami et al., 2019) considers graph node embedding over a homogeneous negative-curvature Riemannian manifold called hyperboloid manifold. In contrast, we generalize the embedding of nodes to an arbitrary pseudo-Riemannian manifold through the geodesic equation (8) with a **learnable** metric $g$ that derives the local graph geometry from the nodes and their neighbors.

## 3.3 FROM GEODESICS TO GENERAL HAMILTONIAN ORBITS

The geodesic curves and the derived exponential map essentially come from (5) with $L$ in (7) specified from the curve energy (4). However, the curves derived from this specific action may potentially sacrifice efficacy for the graph node embedding task since we do not know what is a reasonable action formulation that guides the evolution of the node feature in this task. Therefore, we follow the principle of stationary action but consider a **learnable action** that is more flexible than the length or energy of the curve. To better model the conserved quantity during the feature evolution, we reformulate the Lagrange equation to the Hamilton equation. This is our **Motivation II**.

**Hamiltonian function and equation.** The Hamiltonian orbit $(q(t), p(t))$ is given by the following *Hamiltonian equation* with a *Hamiltonian function $H$*:

$$\dot{q}^i = \frac{\partial H}{\partial p_i}, \quad \dot{p}_i = -\frac{\partial H}{\partial q^i}, \tag{10}$$

where $q$ is the local chart coordinate on the manifold while $p$ can be interpreted as a vector of **generalized momenta** in the cotangent vector space. In classical mechanics, the $2d$-dimensional pair $(p, q)$ is called *phase space* coordinates that fully specify the state of a dynamic system. Later, we consider the node feature evolution following the trajectory specified by the phase space coordinates.

**Hamiltonian function vs. Lagrangian function.** The Hamiltonian function can be taken as the Legendre transform of the Lagrangian function:

$$H(q, p) = p_i\dot{q}^i - L(q, \dot{q}) \text{ with } \dot{q} = \dot{q}(p) \text{ such that } p = \frac{\partial L}{\partial \dot{q}}. \tag{11}$$

If $H$ is restricted to strictly convex functions, the Hamiltonian formalism is equivalent to a Lagrangian formalism (De León & Rodrigues, 2011).

**Geodesic equation reformulated as Hamiltonian function.** If $H$ is set as

$$H(q, p) = \frac{1}{2}g^{ij}(q)p_ip_j, \tag{12}$$

where $[g^{ij}]$ denotes the inverse matrix of the matrix $[g_{ij}]$, we have the following Hamiltonian function:

$$\dot{q}^i = g^{ij}p_j, \quad \dot{p}_i = -\frac{1}{2}\partial_i g^{jk}p_j p_k. \tag{13}$$

The Hamiltonian orbit $(p(t), q(t))$, as the solution of (13), gives us again the *geodesic curves* $p(t)$ on the manifold $M$ if we only look at the first $d$-dimensional coordinates.

**Theorem 1** (Conservation of energy (Da Silva & Da Salva, 2008))**.** $H(p(t), q(t))$ *is constant along the Hamiltonian orbit as solutions of* (10).

In physics, $H$ typically represents the total energy of the system, and Theorem 1 indicates the time-evolution of the system follows the law of conservation of energy.

## 4   PROPOSED FRAMEWORK

We consider an undirected graph $\mathcal{G} = (\mathcal{V}, \mathcal{E})$ consisting of a finite set $\mathcal{V}$ of vertices, together with a subset $\mathcal{E} \subset \mathcal{V} \times \mathcal{V}$ of edges. Since the input node features in most datasets are sparse, a fully connected (FC) layer is first applied to compress the raw input node features. Let $^n q$ be the $d$-dimensional compressed node feature for node $n$ after the FC layer.[2] However, empirical experiments (see "MLP" results in Section 5.1) indicate that for the graph node embedding task, such simple raw compressing without any consideration of the graph topology does not render a good embedding. Further graph neural network architecture is thus required to update the node embedding.

We consider the node features $\{^n q\}_{n \in \mathcal{V}}$ to be located in a local chart of an embedding manifold $M$ and take the node features as the chart coordinate representations for points on the manifold. In **Motivations I and II** in Section 3, we have provided the rationale **for generalizing graph node embedding from the hyperbolic exponential map to the Riemannian geodesic, and further to the Hamiltonian orbit.** To enforce the graph node feature update on the manifold with well-adapted learnable local geometry, we make use of the concepts from Section 3.

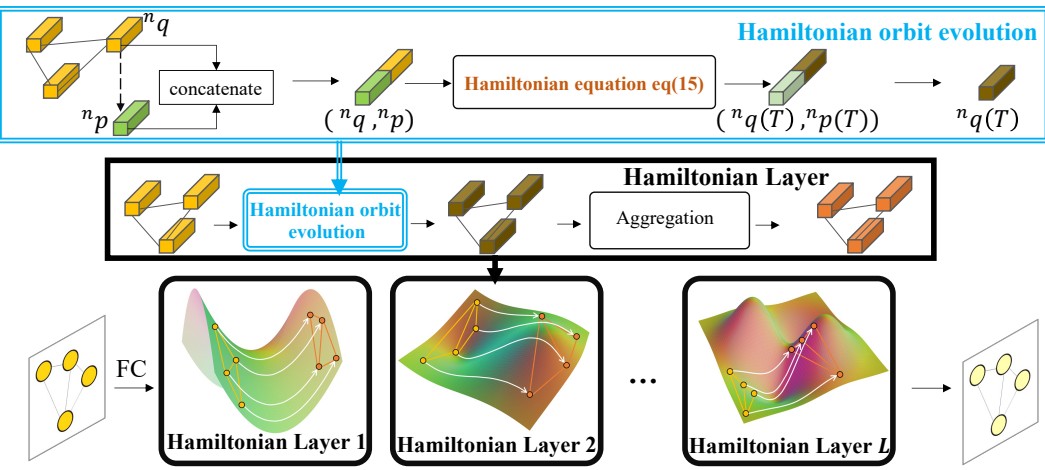

Figure 1: HamGNN architecture: in each layer, each node is assigned a learnable "momentum" vector $^n p$ (cf. (14)) at time $t = 0$, which initializes the evolution of the node feature. The node features evolve on a manifold following (15) to $(^n q(T), ^n p(T))$ at the time $t = T$. We only take $^n q(T)$ as the embedding and input it to the next layer. After $L$ layers, we take $^n q^{(L)}(T)$ as the final node embedding.

### 4.1   MODEL ARCHITECTURE

**Node feature evolution along Hamiltonian orbits in a Hamiltonian layer.** As introduced in Section 3.3, the $2d$-dimensional *phase space* coordinates $(p, q)$ fully specify a system's state. Consequently, for the node feature as a point on the manifold $M$, we associate to each point $^n q$ a *learnable momentum vector* $^n p$ as an external force to enable the node feature to evolve along Hamiltonian orbits on the manifold. More specifically, we set

$$^n p = Q_{\text{net}}(^n q) \tag{14}$$

---

[2]We put the node index to the left of the variable to distinguish it from the manifold dimension index.

where $Q_{\text{net}}$ is instantiated by an FC layer. We consider a learnable Hamiltonian function $H_{\text{net}}$[3] that specifies the node feature evolution trajectory in the phase space by the *Hamiltonian equation*

$$\dot{q}^i = \frac{\partial H_{\text{net}}}{\partial p_i}, \quad \dot{p}_i = -\frac{\partial H_{\text{net}}}{\partial q^i} \tag{15}$$

with learnable Hamiltonian energy function

$$H_{\text{net}} : (q, p) \mapsto \mathbb{R}. \tag{16}$$

The node features are updated along the Hamiltonian orbits which are curves starting from each node $(^nq, {}^np)$ at $t = 0$. In other words, they are the solution of (15) with the initial conditions $(^nq(0), {}^np(0)) = (^nq, {}^np)$ at $t = 0$. The solution of (15) on the phase space for each node $n \in \mathcal{V}$ at time $T$ is given by the differential equation solver (Chen et al., 2018a), and denoted by $(^nq(T), {}^n p(T))$. The canonical projection $\pi(^nq(T), {}^np(T)) = {}^nq(T)$ is taken to obtain the node feature positions on the manifold at time $T$. The aforementioned operations are performed within one layer, and we call it the *Hamiltonian layer*.

**Neighborhood Aggregation.** After the node features update along the Hamiltonian orbits, we perform neighborhood aggregation on the features $\{^nq^{(\ell)}(T)\}_{n \in \mathcal{V}}$, where $\ell$ indicates the $\ell$-th layer. Let $\mathcal{N}(n) = \{m : (n, m) \in \mathcal{E}\}$ denote the set of neighbors of node $n \in \mathcal{V}$. We only perform a simple yet efficient aggregation for node $n$ as follows:

$$^nq^{(\ell+1)} = {}^nq^{(\ell)}(T) + \frac{1}{|\mathcal{N}(n)|} \sum_{m \in \mathcal{N}(n)} {}^mq^{(\ell)}(T). \tag{17}$$

**Layer stacking for local geometry learning.** We stack up multiple Hamiltonian layers with neighborhood aggregation in between them. We first give an intuitive explanation for the case $H_{\text{net}}$ is set as (12) where **a learnable metric $g_{\text{net}}$ for the manifold is involved** (see Section 4.2.1 for more details) and the features are evolved following the geodesic curves with minimal length (see Section 3). Within each Hamiltonian layer, the metric $g_{\text{net}}$ that is instantiated by a smooth FC layer only depends on the local node position on a pseudo-Riemannian manifold that varies from point to point. Note that with layer stacking, these features contain information aggregated from their neighbors. The metric $g_{\text{net}}$, therefore, learns from the graph topology, and each node is embedded with a local geometry that depends on its neighbors. In contrast, (Chami et al., 2019) considers graph node embedding using geodesic curves over a homogeneous negative-curvature hyperboloid manifold without adjustment of the local geometry.

At the beginning of Section 4, we have assumed the node features $\{^nq\}_n$ to be located in a local chart of a preliminary embedding manifold $M$. **The basic philosophy is that the embedding manifold evolves with a metric structure that adapts successively with neighborhood aggregation along multiple layers, whereas each node's features evolve to the most appropriate position as the embedding on the manifold along the curves.** For a general learnable $H_{\text{net}}$, the Hamiltonian orbit that starts from one node has aggregated information from its neighbors, which guides the learning of the curve that the node will be evolved along on the manifold. Therefore, each node is embedded into a manifold with good adaptation to the underlying geometry of any given graph dataset even if it has diverse geometries.

**Conservation of $H_{\text{net}}$.** From Theorem 1, the feature updating through the orbit indicates that the $H_{\text{net}}$ is conserved along the curve.

**Model summary.** Our model is called *HamGNN* as we use Hamiltonian orbits for node feature updating on the manifold. We summarize the HamGNN model architecture in Fig. 1. The forms of the Hamiltonian function $H_{\text{net}}$ are given in Section 4.2.

## 4.2 DIFFERENT HAMILTONIAN ORBITS

We next propose different forms for $H_{\text{net}}$ from which the corresponding Hamiltonian orbit and its variations are obtained in our GNN model. The node features are updated along the Hamiltonian orbits, which are curves starting from each node $(^nq, {}^np)$ at $t = 0$.

### 4.2.1 LEARNABLE METRIC $g_{\text{net}}$

In this subsection, we consider node embedding onto a pseudo-Riemannian and set $H_{\text{net}}$ as (12) where **a learnable metric $g_{\text{net}}$ for the manifold is involved**. Within each Hamiltonian layer, the

---

[3]The subscript net of a function $F_{\text{net}}$ indicates that the function $F$ is parameterized by a neural network.

metric $g_{\text{net}}$ instantiated by a smooth FC layer only depends on the local node position on the pseudo-Riemannian manifold that varies from point to point. The output of $g_{\text{net}}$ at position $q$ represents the inverse metric local representation $[g^{ij}]$. However, from (12), the space complexity is order $d^3$ due to the partial derivative of $g$'s output being a $d \times d$ matrix. We therefore only consider *diagonal metrics* to mitigate the space complexity. More specifically, we now define

$$g_{\text{net}}(q) = \text{diag}([\underbrace{-1, \ldots, -1}_{r}, \underbrace{1, \ldots, 1}_{s}] \odot h_{\text{net}}(q)) \tag{18}$$

where $h_{\text{net}} : \mathbb{R}^d \to \mathbb{R}^d$ consists of non-linear trainable layers and $\odot$ denotes element-wise multiplication. To ensure non-degeneracy of the metric, the output of $h_{\text{net}}$ is set to be away from 0 with the final activation function of it being *strictly positive*. The vector $[-1, \ldots, -1, 1, \ldots, 1]$ controls the signature $(r, s)$ of the metric $g$ with $r + s = d$, where $r$ and $s$ are the number of $-1$s and $1$s, respectively. The signature of the metric is set to be a hyperparameter. According to (13), we have

$$\dot{q}^i = g_{\text{net}}^{ij} p_j, \quad \dot{p}_i = -\frac{1}{2} \partial_i g_{\text{net}}^{jk} p_j p_k. \tag{19}$$

Intuitively, the node features evolve through the "shortest" curves on the manifold. The exponential map used in hyperbolic GNNs (Chami et al., 2019) is essentially the *geodesic* curve on a hyperbolic manifold with an explicit formulation due to the hyperbolic assumption. We do not enforce any assumption here and let the model learn the embedding geometry.

### 4.2.2 LEARNABLE $H_{\text{net}}$

Different from Section 4.2.1 where $H$ is set as (12) with a pseudo-Riemannian metric, we choose a more flexible $H$ instantiated by an FC layer and consider the Hamiltonian function:

$$\dot{q}^i = \frac{\partial H_{\text{net}}}{\partial p_i}, \quad \dot{p}_i = -\frac{\partial H_{\text{net}}}{\partial q^i}. \tag{20}$$

### 4.2.3 LEARNABLE CONVEX $H_{\text{net}}$

As discussed in Section 3.3, if $H_{\text{net}}$ is restricted to strictly convex functions, the Hamiltonian formalism can degenerate to a *Lagrangian formalism* through the Legendre transformation (11). We take the following restricted Hamiltonian equation

$$\dot{q}^i = \frac{\partial H_{\text{net}}}{\partial p_i}, \quad \dot{p}_i = -\frac{\partial H_{\text{net}}}{\partial q^i}, \quad \text{s. t. } H_{\text{net}} \text{ is convex}, \tag{21}$$

where a stationary action in (5) is achieved. To guarantee that $H_{\text{net}}$ is convex, we follow the work in Amos et al. (2017) to set non-negative layer weights from the second layer in $H_{\text{net}}$, and all activation functions in $H_{\text{net}}$ to be convex and non-decreasing. This network design is shown to be able to approximate any convex functions in Chen et al. (2018b).

### 4.2.4 LEARNABLE $H_{\text{net}}$ WITH RELAXATION

Different from Section 4.2.2, we enforce additional system biases along the curve as follows:

$$\dot{q}^i = \frac{\partial H_{\text{net}}}{\partial p_i}, \quad \dot{p}_i = -\frac{\partial H_{\text{net}}}{\partial q^i} + f_{\text{net}}(q). \tag{22}$$

Instead of keeping the energy during the feature update along the Hamiltonian orbit, we now also include an additional energy term during the node feature update.

### 4.2.5 LEARNABLE $H_{\text{net}}$ WITH A FLEXIBLE SYMPLECTIC FORM

Hamiltonian equations have a coordinate-free representation using the symplectic 2-form. We present a self-contained review of differential geometry that is related to symplectic 2-form and Hamiltonian equations in Appendix A. The chart coordinate representation $(q, p)$ may not be the Darboux coordinate system for the symplectic 2-form. Even if our learnable $H_{\text{net}}$ may be able to learn the energy representation under the chosen chart coordinate system, we consider a learnable symplectic 2-form to act in concert with $H_{\text{net}}$. More specifically, following (Chen et al., 2021), we have

$$\theta_{\text{net}}^1 = f_{i,\text{net}} \mathrm{d} q^i,$$

where $f_{\text{net}} : M \to \mathbb{R}^d$ is the output's $i$-th component of the neural network parameterized function from which the Hamiltonian orbit is given by (Chen et al., 2021) as follows:

$$(\dot{q}^i, \dot{p}^i) = W^{-1}(q, p) \nabla H_{\text{net}}(q, p), \tag{23}$$

where the skew-symmetric $2d \times 2d$ matrix $W$, whose elements are written in terms of $(\partial_i f_{j,\text{net}} - \partial_j f_{i,\text{net}})$, is given in (50) due to space constraints.

## 5 EXPERIMENTS

In this section, we implement the proposed HamGNNs with different settings as shown in Section 4.2 and Appendix B. We select **datasets with various geometry** including the three citation networks Cora (McCallum et al., 2004), Citeseer (Sen et al., 2008), Pubmed (Namata et al., 2012), and two low hyperbolicity datasets (Chami et al., 2019), named Disease and Airport as the benchmark datasets. To fairly compare the performance of the proposed HamGNN, we select several popular GNN models as the baseline, including the Euclidean GNNs: GCN (Kipf & Welling, 2017a), GAT (Veličković et al., 2018), SAGE (Hamilton et al., 2017), and SGC (Wu et al., 2019); the hyperbolic GNNs (Chami et al., 2019; Liu et al., 2019): HGNN, HGCN, and HGAT, and also GIL (Zhu et al., 2020b) which learns a weighting between Euclidean and hyperbolic space embeddings; the Graph Neural diffusion GNNs: GRAND (Chamberlain et al., 2021b), GraphCON Rusch et al. (2022) and LGCN (Zhang et al., 2021b). The MLP baseline does not utilize the graph topology information. To further demonstrate the advantage of HamGNN, we also include one vanilla ODE system, whose formulation is given in Appendix D.1 without the crucial Hamiltonian layer. Due to space constraints, we refer the readers to Appendix C for the datasets and implementation details. **We stress at the outset that we do not aim to outperform all the baselines or other general-purpose GNN models on specific datasets. Instead, our objective is to design a new node embedding strategy that can automatically learn, without extensive tuning, the underlying geometry of any given graph dataset even if it has diverse geometries.** To demonstrate that HamGNN can automatically adapt well, we include two graph node embedding downstream tasks, including node classification and link prediction. Due to space constraints, we refer the readers to Appendix C for the datasets and implementation details.

| Method | Disease | Airport | Pubmed | Citeseer | Cora |
|---|---|---|---|---|---|
| MLP | 50.00±0.00 | 76.96±1.77 | 71.95±1.38 | 58.10±1.87 | 57.15±1.15 |
| HNN (Ganea et al., 2018) | 56.40±6.32 | 80.49±1.54 | 71.60±0.47 | 55.13±2.04 | 58.03±0.55 |
| GCN (Kipf & Welling, 2017a) | 81.10±1.33 | 82.25±0.56 | 77.83±0.77 | 71.78±0.34 | 80.29±2.29 |
| GAT (Veličković et al., 2018) | 87.01±2.77 | 92.99±0.83 | 77.58±0.81 | 68.07±1.31 | 80.33±0.61 |
| SAGE (Hamilton et al., 2017) | 81.60±7.68 | 81.97±0.85 | 77.63±0.15 | 65.90±2.32 | 74.50±0.88 |
| SGC (Wu et al., 2019) | 82.78±0.93 | 81.40±2.21 | 76.83±1.11 | 70.88±1.32 | 81.98±1.71 |
| HGNN (Liu et al., 2019) | 80.51±5.70 | 84.54±0.72 | 76.65±1.38 | 69.43±0.99 | 79.53±0.98 |
| HGCN (Chami et al., 2019) | 89.87±1.13 | 85.35±0.65 | 76.38±0.81 | 65.80±2.04 | 78.70±0.96 |
| HGAT (Zhang et al., 2021a) | 88.68±3.36 | 87.50±0.99 | 78.00±0.50 | 69.20±0.96 | 80.88±0.75 |
| GIL (Zhu et al., 2020b) | 90.78±0.45 | 91.52±1.74 | 77.76±0.57 | 71.10±1.24 | 82.10±1.12 |
| Vanilla ODE in (53) | 71.81±18.85 | 90.34±0.67 | 73.30±3.31 | 56.60±1.26 | 68.38±1.18 |
| GRAND (Chamberlain et al., 2021b) | 74.52±3.37 | 60.02±1.55 | **79.32±0.51** | 71.76±0.79 | **82.80±0.92** |
| GraphCON Rusch et al. (2022) | 87.50±4.06 | 68.61±2.10 | 78.80±0.97 | 71.33±0.83 | _82.49±1.08_ |
| LGCN (Zhang et al., 2021b) | 88.47±1.80 | 88.22±0.18 | 77.35±1.38 | 68.08±1.98 | 80.60±0.92 |
| HamGNN (19) | _91.26±1.40_ | 95.50±0.48 | 78.08±0.48 | 70.12±0.86 | 82.16±0.80 |
| HamGNN (20) | 88.74±1.17 | 95.11±0.40 | 78.18±0.54 | 71.48±1.42 | 81.52±1.27 |
| HamGNN (21) | 84.57±5.78 | 93.28±1.40 | _78.83±0.46_ | _72.00±0.73_ | 81.84±0.88 |
| HamGNN (22) | 87.48±5.90 | 95.46±0.68 | 78.30±0.34 | **72.38±0.85** | 81.56±0.97 |
| HamGNN (23) | 88.35±1.51 | 93.66±0.16 | 78.60±0.32 | 71.52±1.41 | 81.24±0.59 |
| HamGNN (51) | 91.18±0.99 | _95.80±0.19_ | 77.90±0.49 | 69.18±1.63 | 80.90±0.35 |
| HamGNN (52) | **91.50±2.07** | **95.99±0.13** | 78.26±0.64 | 69.10±1.95 | 80.10±1.56 |

Table 1: Node classification accuracy(%). The best and the second-best result for each criterion are highlighted in **bold** and _underlined and italic_ respectively.

### 5.1 PERFORMANCE RESULTS AND ABLATION STUDIES

**Node classification.** The node classification performance on the benchmark datasets using the baseline models and the proposed HamGNNs with different $H_{net}$s is shown in Table 1. We observe that HamGNN adapts well to all datasets with various geometry. As an illustration, HamGNN (19) achieves the third best performance on the Cora dataset, which is Euclidean in nature. According to (Chami et al., 2019; Liu et al., 2019), the Airport dataset has a tree-like structure that cannot be properly embedded into Euclidean spaces. We observe that HamGNN (19) has a well-adapted performance that beats all the baselines on this dataset due to the learnable embedding manifold without many constraints. As discussed in Section 4.1, HamGNN with the learnable metric structure $g_{net}$ or other general $H_{net}$ learns the graph local geometry successively with neighborhood aggregation. The superior performance of HamGNNs on all the datasets over the other baselines has demonstrated the advantage of including Hamiltonian orbits on the manifold to embed the node features. Furthermore, the comparison between HamGNNs with the GNN with a vanilla ODE in (53) without any Hamiltonian mechanism indicates the indispensability of the Hamiltonian layer.
**Link prediction.** In Table 2, we report the averaged ROC for the link prediction task. We observe

HamGNN adapts well to all datasets and is the best performer on the Airport, Citeseer, and Cora.

| Method | Disease | Airport | Pubmed | Citeseer | Cora |
|---|---|---|---|---|---|
| MLP | 83.37±5.04 | 87.04±0.56 | 88.69±1.59 | 89.65±1.00 | 91.07±0.56 |
| HNN (Ganea et al., 2018) | 81.37±8.78 | 86.06±2.08 | 94.69±0.25 | 89.83±0.39 | 92.83±0.76 |
| GCN (Kipf & Welling, 2017a) | 60.38±2.51 | 90.97±0.65 | 91.37±0.09 | 93.20±0.28 | 92.89±0.77 |
| GAT (Veličković et al., 2018) | 62.03±1.58 | 91.05±0.83 | 91.03±0.67 | 93.83±0.65 | 93.34±0.50 |
| SAGE (Hamilton et al., 2017) | 68.02±0.43 | 91.40±0.88 | 93.61±0.26 | 93.37±0.88 | 92.94±0.40 |
| SGC (Wu et al., 2019) | 59.83±4.01 | 89.72±0.82 | 92.16±0.13 | 94.78±0.77 | 93.15±0.22 |
| HGNN (Liu et al., 2019) | 60.20±1.14 | 92.46±0.20 | 93.09±0.09 | 90.35±0.57 | 92.05±0.33 |
| HGCN (Chami et al., 2019) | 78.09±2.79 | 94.28±0.20 | _96.79±0.01_ | 93.60±0.14 | 94.10±0.05 |
| HGAT (Zhang et al., 2021a) | 76.32±3.41 | 94.64±0.51 | **96.86±0.03** | 93.45±0.25 | 94.96±0.36 |
| GIL (Zhu et al., 2020b) | **99.97±0.08** | _97.92±2.64_ | 91.22±3.25 | _95.99±8.89_ | _97.78±2.31_ |
| HamGNN | _99.73±0.26_ | **99.99±0.01** | 92.15±0.30 | **99.99±0.00** | **98.20±1.73** |

Table 2: Link prediction ROC(%). The best and the second-best result for each criterion are highlighted in **bold** and _underlined and italic_ respectively.

**Comparison between different $H_{net}$.** We compare HamGNNs with different $H_{net}$s as elaborated in Section 4.2 on the node classification task. In Section 3.3, we argue that the geodesic curves derived from the action of curve length (3) may potentially sacrifice efficacy for the graph node embedding task since we do not know what is a reasonable action formulation that guides the evolution of the node feature in this task. We therefore also include a more flexible $H_{net}$ in (20) with other variations, e.g., (21) to (23). From Table 1, we observe that the basic (20) has good adaptations for node embedding of various datasets. Flexibility in (20) helps the node classification task slightly for the Citeseer dataset, but the improvement is not significant. Other variants of $H_{net}$ also present good adaptations for all the datasets. This observation indicates that good geometry adaptations may come from the HamGNN model architecture with the Hamiltonian orbit evolution. This is further verified by the node classification performance from vanilla ODE in (53), which is designed without the philosophy of Hamiltonian mechanics. Vanilla ODE (53) does not have a well-adapted node embedding performance for datasets with various geometries.

We then compare the HamGNN using (20) and (23). The difference between those two settings is that the symplectic form $\omega^2$ in (20) is set to be the special Poincaré 2-form while in (23), the symplectic form is a learnable one. We however observe that the two HamGNNs achieve similar performance and the more flexible symplectic form does not improve the model performance. This may be because of the fundamental Darboux theorem Theorem 3 in symplectic geometry which states that we can always find a Darboux coordinate system to give any symplectic form the _Poincaré 2-form_. The feature compressing FC may have the network capacity to approximate the Darboux coordinate map while the flexible learnable $H_{net}$ also has the network capacity to get the energy representation under the chosen chart coordinate system.

| Dataset | Models | 3 layers | 5 layers | 10 layers | 20 layers |
|---|---|---|---|---|---|
| | GCN | 80.29±2.29 | 69.87±1.12 | 26.50±4.68 | 23.97±5.42 |
| Cora | HGCN | 78.70±0.96 | 38.13±6.20 | 31.90±0.00 | 26.23±9.87 |
| | HamGNN (21) | **81.84±0.88** | 81.08±0.16 | **81.40±0.44** | **80.58±0.30** |

Table 3: Node classification accuracy(%) when increasing the number of layers on the Cora dataset.

## 5.2 OBSERVATION OF RESILIENCE TO OVER-SMOOTHING

As a side benefit of HamGNN, we observe from Table 3 that if more Hamiltonian layers are stacked, HamGNN still retains its node classification ability, while the classification accuracies of other GNNs decrease significantly. This is known as the _over-smoothing_ problem (Chen et al., 2020a) in GNNs. One reason for HamGNN's stability against over-smoothing is that the node feature energy indicated by the Hamiltonian $H_{net}$ is fixed during the feature update along the Hamiltonian orbit.

**More Ablation Studies and Experiments.** See Appendix D where we include experimental results on heterophilic graph datasets, more empirical analysis of oversmoothness, etc.

## 6 CONCLUSION

In this paper, we have designed a new node embedding strategy from Hamiltonian orbits that can automatically learn, without extensive tuning, the underlying geometry of any given graph dataset even if it has diverse geometries. We demonstrate empirically that our approach adapts better than popular state-of-the-art graph node embedding GNNs to different types of graph datasets via two graph node embedding downstream tasks, including node classification and link prediction. From experiments, we observe that the over-smoothing problem can be mitigated if the node features evolve through the Hamiltonian orbits.

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

## A  DIFFERENTIAL GEOMETRY AND HAMILTONIAN SYSTEM

In this supplementary material, we review some concepts from a differential geometry perspective. We hope that this overview makes the paper more accessible for readers from the graph learning community.

### A.1  MANIFOLD, BUNDLES, AND FIELDS

Roughly speaking, a manifold is a topological space that *locally* looks like Euclidean space. More strictly speaking, a topological space $(M, \mathcal{O})$, where $\mathcal{O}$ is the collection of open sets on space $M$, is called a $d$-dimensional manifold if for every point $x \in M$, we can find an open neighborhood $U \in \mathcal{O}$ for $x$ and a coordinate map

$$q : U \to q(U) \subseteq \mathbb{R}^d$$

that is a *homeomorphism*, where $\mathbb{R}^d$ is the $d$-dimensional Euclidean space with the standard topology. $(U, q)$ is called a *chart* of the manifold, which gives us a numerical representation for a local area in $M$. In this work, we only consider smooth manifolds (Lee, 2013) that any two overlapped charts are smoothly compatible. The set of all smooth functions from $M$ to $\mathbb{R}$ is denoted as $\mathcal{C}^\infty(M)$. On top of the smooth manifolds, we can define other related manifolds like the tangent or cotangent bundles and the more general tensor bundles. From bundles, we can define the vector or covector fields and the more general tensor fields. In this work, we mainly consider the manifold with the 2-forms that are special $(0, 2)$ smooth tensor fields with antisymmetric constraints. More specifically, we mainly consider the *symplectic form* (Lee, 2013) with some other light shed on the metric tensor which is another type of $(0, 2)$ smooth tensor field.

**Definition 1** (tangent vector and tangent space). *Let $\gamma : I \to M$ be a smooth curve through $x \in M$ s.t. $\gamma(0) = x$ and $I$ is an interval neighborhood of $0$. The* tangent vector *is a directional derivative operator at $x$ along $\gamma$ that is the* linear map

$$v_{\gamma, x} : \mathcal{C}^\infty(M) \xrightarrow{\sim} \mathbb{R}$$
$$f \mapsto (f \circ \gamma)'(0).$$

*We also call the directional derivative operator as "velocity" of $\gamma$ at $0$ and denote it as $\dot{\gamma}(0)$. (More generally, the 'velocity' of $\gamma$ at $t$ is denoted at $\dot{\gamma}(t)$ which is a tangent vector at point $\gamma(t) \in M$ from*

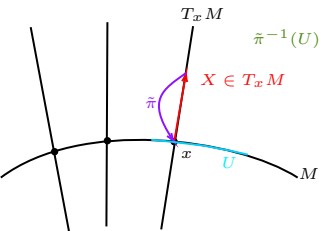

Figure 2: Visualization of a 1-dimensional manifold and its tangent bundle.

*the curve reparametrization trick (Fecko, 2006)). Correspondingly, the* tangent space *to $M$ at $x$ is the* vector space *over $\mathbb{R}$ with the underlying set*

$$T_x M := \{v_{\gamma,x} \mid \gamma \text{ is a smooth curve and } \gamma(0) = x\}. \tag{24}$$

*Note for $f \in \mathcal{C}^\infty(M)$, using the chart $(U, q)$ with $x \in U$, we have the local representation*

$$
\begin{aligned}
v_{\gamma,x}(f) &:= (f \circ \gamma)'(0) \\
&= (f \circ q^{-1} \circ q \circ \gamma)'(0) \\
&= (q^i \circ \gamma)'(0) \cdot \partial_i \big(f \circ q^{-1}\big)\big|_{q(x)},
\end{aligned}
$$

*where $q^i$ is the $i$-th component of $q$, $\circ$ is the function composition and $(\cdot)|_{q(x)}$ means evaluating $(\cdot)$ at $q(x)$. Therefore for a local chart around $x$, we have a basis of $T_x M$ as*

$$\left\{ \left(\frac{\partial}{\partial q^1}\right)_x, ..., \left(\frac{\partial}{\partial q^d}\right)_x \right\}$$

*and we call it the* chart induced basis, *where $\left(\frac{\partial}{\partial q^1}\right)_x := \partial_i(f \circ q^{-1})|_{q(x)}$.*

**Definition 2** (tangent bundle). *Given a smooth manifold $M$, the* tangent bundle *of $M$ is the* disjoint union *of all the tangent spaces to $M$, i.e., we have*

$$TM := \coprod_{x \in M} T_x M, \tag{25}$$

*equipped with the* canonical projection map

$$
\begin{aligned}
\tilde{\pi} : \ &TM \to M \\
&X \mapsto \tilde{\pi}(X),
\end{aligned}
$$

*where $\tilde{\pi}(X)$ sends each vector in $T_x M$ to the point $x$ at which it is tangent, and $\coprod$ is the disjoint union. Furthermore, the tangent bundle[4] is a $2d$-dimensional manifold. If $X \in \tilde{\pi}^{-1}(U) \subseteq TM$ with a local chart $(U, q)$ on $M$ s.t. $x \in U$, then $X \in T_{\tilde{\pi}(X)} M$ from the definition. Since $\tilde{\pi}(X) \in U$, $X$ can be written in terms of the* chart induced basis*:*

$$X = \tilde{p}^i(X) \left(\frac{\partial}{\partial q^i}\right)_{\tilde{\pi}(X)}, \tag{26}$$

*where $\tilde{p}^1, \ldots, \tilde{p}^d$ are smooth scalar functions. We can then define the following map as a local chart for the manifold $TM$ induced from the chart $(U, q)$ on $M$:*

$$
\begin{aligned}
\xi : \ &\tilde{\pi}^{-1}(U) \to q(U) \times \mathbb{R}^d \subseteq \mathbb{R}^{2d} \\
&X \mapsto (q^1(\tilde{\pi}(X)), \ldots, q^d(\tilde{\pi}(X)), \tilde{p}^1(X), \ldots, \tilde{p}^d(X)),
\end{aligned}
$$

*and the topological structure on $TM$ is derived from the initial topology to ensure continuity.*

---

[4]In this paper, we may just use the word "bundle" to indicate the total space in the bundle.

**Definition 3** (vector field). *A vector field on $M$ is a smooth section (Lee, 2013) of the tangent bundle, i.e. a smooth map $\sigma : M \to TM$ such that $\tilde{\pi} \circ \sigma = id_M$, where $id_M$ is the identity map on $M$.*

$$
\begin{array}{c}
TM \\
\sigma \Big\uparrow \Big\downarrow \tilde{\pi} \\
M
\end{array}
\tag{27}
$$

*We denote the set of all vector fields on $M$ by $\Gamma(TM)$:*

$$
\Gamma(TM) := \{\sigma : M \to TM \mid \sigma \text{ is smooth and } \tilde{\pi} \circ \sigma = id_M\}. \tag{28}
$$

**Definition 4** (cotangent vector, cotangent bundle, dual basis, and covector field). *For the vector space $T_xM$, a continuous linear functional from $T_xM$ to $\mathbb{R}$ is called a cotangent vector at $x$. The set of all such linear maps is denoted as $T_x^*M$ which is the dual vector space of $T_xM$. For $f \in \mathcal{C}^\infty(M)$, at each point $x$, we define the following linear operator in $T_x^*M$*

$$
(\mathrm{d}f)_x : T_xM \to \mathbb{R}
$$
$$
X_x \mapsto (\mathrm{d}f)_x(X_x) := X_x(f).
$$

*Given a chart $(U, q)$ with $x \in U$ and its* chart induced basis*, the* dual basis *for the dual space $T_x^*M$ is the set*

$$
\left\{\left(\mathrm{d}q^1\right)_x, \ldots, \left(\mathrm{d}q^d\right)_x\right\},
$$

*where we have $(\mathrm{d}q^a)_x \left(\left(\frac{\partial}{\partial q^b}\right)_x\right) = \left(\frac{\partial}{\partial q^b}\right)_x (q^a) = \delta_b^a$ with $\delta_b^a = 1$ iff $a = b$ and $\delta_b^a = 0$ otherwise. We call it the* chart induced dual basis *. Analogous to the above definition of the vector field, we can define the cotangent bundle of $M$ as*

$$
T^*M := \coprod_{x \in M} T_x^*M \tag{29}
$$

*which is again a $2d$-dimensional manifold equipped with the* canonical projection map

$$
\pi : T^*M \to M
$$
$$
\omega \mapsto \pi(\omega), \tag{30}
$$

*where $\pi(\omega)$ sends each vector in $T_x^*M$ to the point $x$ at which it is cotangent. If $\omega \in \pi^{-1}(U) \subseteq T^*M$ with a local chart $(U, q)$ s.t. $x \in U$, then $\omega \in T_{\pi(\omega)}^*M$ from the definition. Since $\pi(\omega) \in U$, $\omega$ can be written in terms of the* chart induced dual basis*:*

$$
\omega = p_i(\omega)(\mathrm{d}q^i)_{\pi(\omega)}, \tag{31}
$$

*where $p_1, \ldots, p_d$ are smooth scalar functions. We can then define the following map as a local chart for manifold $T^*M$ induced from the chart $(U, q)$ on $M$:*

$$
\xi : \pi^{-1}(U) \to q(U) \times \mathbb{R}^d \subseteq \mathbb{R}^{2d}
$$
$$
\omega \mapsto (q^1(\pi(\omega)), \ldots, q^d(\pi(\omega)), p_1(\omega), \ldots, p_d(\omega)),
$$

*and the topological structure on $T^*M$ is derived from the initial topology to ensure continuity. The covector fields are smooth sections of $T^*M$. The set of all covector fields is denoted as $\Gamma(T^*M)$.*

**Definition 5** (tensor field and 2-form). *The tensor field can be defined using the smooth sections on tensor bundles analogously to the vector fields or the covector fields. We refer readers to (Lee, 2013) for more details. Here, instead of a rigorous definition, we show some basic properties of the tensor fields. For a $(r, s)$ tensor field $\tau$, at $x \in M$, it is a multilinear map*

$$
\tau_x : \underbrace{T_x^*M \times \cdots \times T_x^*M}_{r \text{ copies}} \times \underbrace{T_xM \times \cdots \times T_xM}_{s \text{ copies}} \to \mathbb{R}.
$$

*The differential $k$-form $\omega$ is the $(0, k)$ tensor field that admits alternating (Lee, 2013). Specifically, for the 2-form $\omega$, at each point $x$, $\omega_x$ is a antisymmetric $(0, 2)$ tensor*

$$
\omega_x : T_xM \times T_xM \to \mathbb{R} \text{ s.t. } \omega_x(X_1, X_2) = -\omega_x(X_2, X_1) \,\forall X_1, X_2 \in T_xM \tag{32}
$$

*which in other words, $\omega$ satisfies*

$$\omega : \Gamma(TM) \times \Gamma(TM) \to \mathcal{C}^\infty(M) \text{ s.t. } \omega(X,Y) = -\omega(Y,X) \,\forall\, X, Y \in \Gamma(TM) \qquad (33)$$

*In local chart representation, we have that every $k$-form $\omega$ can be expressed* locally *on $U$ as*

$$\omega = \omega_{a_1 \cdots a_k} \, \mathrm{d}x^{a_1} \wedge \cdots \wedge \mathrm{d}x^{a_k}, \qquad (34)$$

*where $\omega_{a_1 \cdots a_k} \in \mathcal{C}^\infty(U)$, $1 \le a_1 < \cdots < a_k \le \dim M$ are increasing sequences and $\mathrm{d}x^{a_1} \wedge \cdots \wedge \mathrm{d}x^{a_k}$ is the wedge product (Lee, 2013). Here we could abstractly view the set $\{\mathrm{d}x^{a_1} \wedge \cdots \wedge \mathrm{d}x^{a_k}\}_{a_1,\ldots,a_k}$, with $a_i$ enumerated from 1 to d, abstractly as a basis without more illustrations of the wedge product (We refer readers to (Lee, 2013) for more details).*

**Definition 6** (integral curve). *Given a vector field $X$ on $M$, an integral curve of $X$ is a differentiable curve $\gamma : I \to M$, where $I \subseteq \mathbb{R}$ is an interval, whose velocity at each point is equal to the value of $X$ at that point:*

$$\dot\gamma(t) = X_{\gamma(t)} \quad \text{for all } t \in I. \qquad (35)$$

*If $0 \in J$, the point $\gamma(0)$ is called the starting point of $\gamma$. From Picard's theorem (Hartman, 2002), we know that locally we always have an interval $I$ on which the solution exists and is necessarily unique.*

**Definition 7** (exterior derivative and closed form). *The exterior derivative is a linear operator that maps $k$-forms to $k+1$-forms. In local chart representation, we have that if $\omega$ is a $k$-form on $M$ with the local representation as*

$$\omega = \omega_{a_1 \cdots a_k} \mathrm{d}x^{a_1} \wedge \cdots \wedge \mathrm{d}x^{a_k}. \qquad (36)$$

*Then, we have the exterior derivative*

$$\begin{aligned} \mathrm{d}\omega &= \mathrm{d}\omega_{a_1 \cdots a_k} \wedge \mathrm{d}x^{a_1} \wedge \cdots \wedge \mathrm{d}x^{a_k} \\ &= \partial_b \omega_{a_1 \cdots a_k} \mathrm{d}x^b \wedge \mathrm{d}x^{a_1} \wedge \cdots \wedge \mathrm{d}x^{a_k}. \end{aligned} \qquad (37)$$

*A form $\omega$ is called* closed *if $\mathrm{d}\omega = 0$.*

**Theorem 2.** *From (Rudin et al., 1976; Lee, 2013), we know that*

$$\mathrm{d} \circ \mathrm{d} \equiv 0, \qquad (38)$$

*which is a dual statement that the boundary of the boundary of a manifold is empty from Stokes' theorem.*

## A.2   HAMILTONIAN VECTOR FIELDS ON SYMPLECTIC COTANGENT BUNDLE

**Definition 8** (symplectic vs. Riemannian). *Let $M$ be a smooth manifold.*

1. *symplectic form: A 2-form (so it is antisymmetric) $\omega$ is said to be a symplectic form on $M$ if it is closed, i.e,*

$$\mathrm{d}\omega = 0,$$

   *and it is non-degenerate, i.e,*

$$(\forall\, Y \in \Gamma(TM) : \omega(X,Y) = 0) \Rightarrow X = 0. \qquad (39)$$

2. *metric tensor: A $(0,2)$ tensor field $g$ is said to be a Riemannian metric on $M$ if it is non-degenerate and symmetric at each $x$, i.e.,*

$$g_x : \ T_x M \times T_x M \to \mathbb{R} \text{ s.t. } g_x(X_1, X_2) = g_x(X_2, X_1) \,\forall\, X_1, X_2 \in T_x M \qquad (40)$$

**Remark 1.** *A manifold equipped with a symplectic form $\omega$ is called a symplectic manifold, while a manifold equipped with a metric tensor $g$ is called a pseudo-Riemannian manifold. The Riemannian metric $g$ is a $(0,2)$-tensor field measuring the norms of tangent vectors and the angles between them. To some extent, the "shape structure" of the manifold $M$ is only available if we equipped $M$ with a metric $g$.*

- *From the above definition, we know the symplectic form and the metric tensor are both nondegenerate bilinear $(0, 2)$ tensor fields. One difference is the symplectic form is anti-symmetric while the metric tensor is symmetric. At each point $x \in M$, if we use the local chart coordinate representation, the $(0, 2)$ tensor can be represented as the following matrix multiplication*

$$\tilde{p}^{\mathsf{T}} W_x \tilde{p}$$

  *where $d \times d$ matrix $W$ is symmetric for metric tensor and is skew-symmetric for symplectic form. $\tilde{p}^{\mathsf{T}}$ is the transpose of the velocity representation (which is a numerical vector) in a local chart.*

- *Because of* non-degenerate, *on a symplectic manifold $M$, we can define an isomorphism between $\Gamma(TM)$ and $\Gamma(T^*M)$ by mapping a vector field $X \in \Gamma(TM)$ to a 1-form $\eta_V \in \Gamma(T^*M)$, where*

$$\eta_X(\cdot) := \omega^2(\cdot, X) \tag{41}$$

  *Similarly, on a pseudo-Riemannian manifold, we can define an isomorphism between $\Gamma(TM)$ and $\Gamma(T^*M)$ by mapping a vector field $X \in \Gamma(TM)$ to a 1-form $\in \Gamma(T^*M)$, where*

$$\alpha_g : TM \longrightarrow T^*M. \tag{42}$$

  *In a local chart coordinate representation, $\alpha_g = g_{ij}$ and its inverse $\alpha_g^{-1} = g^{ij}$ with $\sum_{j=1}^m g_{ij} g^{jk} = \delta_i^k$ and $\delta_i^k = 1$ iff $i = k$ and 0 otherwise. Note the components of the metric and the inverse metric are all taken in a given chart without explicitly mentioning them.*

**Definition 9** (Hamiltonian flow and orbit). *For a general symplectic manifold $M$ with a symplectic form $\omega^2$, if we have $H \in \mathcal{C}^\infty(M)$, then $\mathrm{d}H$ is a differential $1$-form on $M$. We define the vector field called the **Hamiltonian flow** $X_H$ **associated to the Hamiltonian** $H$, which satisfies that*

$$\eta_{X_H}(\cdot) = \mathrm{d}H(\cdot).$$

*The integral curves of are called **Hamiltonian orbits of** $H$:*

$$\dot{\gamma}(t) = (X_H)_{\gamma(t)} \quad \text{for all } t \in I. \tag{43}$$

*where $(X_H)_{\gamma(t)}$ is the tangent vector at $\gamma(t) \in M$.*

**Definition 10** (Poincaré 1-form and 2-form). *On the cotangent bundle $T^*M$ of a manifold $M$, we have a natural symplectic form, called the* Poincaré *1-form*

$$\theta^1_{\text{Poincaré}} = p_i \mathrm{d}q^i. \tag{44}$$

*Therefore, by the exterior derivative, we have the* Poincaré *2-form*

$$\omega^2_{\text{Poincaré}} = \mathrm{d}\theta^1 = \mathrm{d}(p_i \mathrm{d}q^i) = \sum_i \mathrm{d}p_i \wedge \mathrm{d}q^i \tag{45}$$

*which is closed from (38). Therefore* Poincaré *2-form* *is a symplectic form on the cotangent bundle and cotangent bundles are the natural **phase spaces** of classical mechanics (De León & Rodrigues, 2011).*

For more general symplectic forms on the cotangent bundle, we can again use (38) to construct the closed 2-form from a 1-form which is potentially symplectic:

**Corollary 1** ((Chen et al., 2021)). *According to (38), on the cotangent bundle $T^*M$ of a manifold $M$, from a $1$-form,*

$$\theta^1 = f_i \mathrm{d}q^i,$$

*we derive a closed $2$-form using the exterior derivative (37), and its local representation is given by the following*

$$\omega^2 = \mathrm{d}\theta^1 = \mathrm{d}(f_i \mathrm{d}q^i) = \sum_{i<j} (\partial_i f_j - \partial_j f_i) \, \mathrm{d}p_i \wedge \mathrm{d}q^j$$

**Remark 2.** *Note, strictly speaking, we only can get the necessary "closed" condition for $\omega^2$ to be potentially symplectic. However, it is enough for our use in our proposed framework.*

We now state one of the most fundamental results in symplectic geometry that links the general symplectic form to the special Poincaré 2-form.

**Theorem 3** (Darboux (Lee, 2013))**.** *Let $(\tilde{M}, \tilde{\omega}^2)$ be a $2d$-dimensional symplectic manifold. For any $x \in \tilde{M}$, there are smooth coordinates $\left(\tilde{q}^1, \ldots, \tilde{q}^d, \tilde{p}^1, \ldots, \tilde{p}^n\right)$ centered at $x$ in which $\omega^2$ has the coordinate representation*

$$\tilde{\omega}^2 = \sum_{i=1}^n \mathrm{d}\tilde{q}^i \wedge \mathrm{d}\tilde{p}^i. \tag{46}$$

*as a Poincaré 2-form, any coordinates satisfying* (46) *are called Darboux or symplectic coordinates.*

**Remark 3.** *Therefore, for any symplectic form $w^2$ on the above cotangent bundle $T^*M$ of $M$, we can always find a symplectic coordinate with the* Poincaré 2-form.

**Corollary 2** ((Da Silva & Da Salva, 2008))**.** *According to* (33)*, from the antisymmetric of the 2-forms, we have*

$$\omega^2(X_H, X_H) = 0 \tag{47}$$

*which implies that hamiltonian vector fields preserve their hamiltonian functions $H$.*

**Remark 4.** *In physics, hamiltonian functions are typically energy functions for a physical system. Corollary 2 indicates if the system updates its status according to the hamiltonian vector fields, the time-evolution of the system follows the law of conservation of energy.*

**Definition 11** (Hamiltonian orbit generated from Poincaré 2-form (Lee, 2013))**.** *The Hamiltonian orbit generated by the Hamiltonian flow $X_H$ on the cotangent bundle equipped with the Poincaré 2-form given in local coordinates $(q, p)$ is given by*

$$\dot{q}^i = \frac{\partial H}{\partial p_i}, \quad \dot{p}_i = -\frac{\partial H}{\partial q^i}. \tag{48}$$

**Definition 12** (cogeodesic orbits)**.** *If additionally $M$ is equipped with a metric tensor g, i.e, if $M$ is a pseudo-Riemannian manifold with metric g, and if we set the hamiltonian $H$ on $T^*M$ as*

$$H(q, p) = \frac{1}{2} g^{ij}(q) p_i p_j,$$

*the Hamiltonian orbit generated from Poincaré 2-form is given by*

$$\dot{q}^i = \frac{\partial H}{\partial p_i} = g^{ij} p_j, \quad \dot{p}_i = -\frac{\partial H}{\partial q^i} = -\frac{1}{2} \partial_i g^{jk} p_j p_k. \tag{49}$$

*It is called the **cogeodesic orbits of** $(M, g)$. The canonical projection of cogeodesic orbits under $\pi$* (30) *is called* geodesic *on the base manifold $M$ which generalizes the notion of a "straight or shortest line" to manifold where the length is measured by the metric tensor.*

## B  SOME FORMULATIONS AND MORE HAMILTONIAN ORBITS

In section, we first present the formulations which are not shown in detail in the main paper due to space constraints. More Hamiltonian-related flows are also presented, which however do not strictly follow the Hamiltonian orbits on the cotangent bundle $T^*M$.

### B.1  $W$ IN SECTION 4.2.5

$$W = \begin{pmatrix} 0 & \partial_1 f_{2,\mathrm{net}} - \partial_2 f_{1,\mathrm{net}} & \partial_1 f_{3,\mathrm{net}} - \partial_3 f_{1,\mathrm{net}} & \cdots \\ \partial_2 f_{1,\mathrm{net}} - \partial_1 f_{2,\mathrm{net}} & 0 & \partial_2 f_{3,\mathrm{net}} - \partial_3 f_{2,\mathrm{net}} & \cdots \\ \partial_3 f_{1,\mathrm{net}} - \partial_1 f_{3,\mathrm{net}} & \partial_3 f_{2,\mathrm{net}} - \partial_2 f_{3,\mathrm{net}} & 0 & \cdots \\ \vdots & \vdots & \vdots & \ddots \end{pmatrix} \tag{50}$$

Similar to Section 4.2.4, we now impose additional system biases along the curve compared to the cogeodesic orbits Section 4.2.1,

$$\dot{q}^i = g^{ij}_{\text{net}} p_j, \quad \dot{p}_i = -\frac{1}{2}\partial_i g^{jk}_{\text{net}} p_j p_k + f_{\text{net}}(q). \tag{51}$$

Therefore, the projection of the curve from (51) now no longer follows the geodesic curve along the base manifold equipped with metric $g_{\text{net}}$.

## B.3 Hamiltonian Relaxation Flow with Higher Dimensional "Momentum"

In the paper main context, we present a new type of Hamiltonian-related flow, which does not strictly follow the Hamiltonian equations. Inspired from the work (Haber & Ruthotto, 2017), we now associate to each node $q \in \mathbb{R}^d$ an additional a *learnable* momentum vector $p \in \mathbb{R}^k$ which however is not strictly a cotangent vector of the manifold if $d \neq k$. We update the node features using the following equations

$$\begin{aligned}
\dot{q} &= \phi\left(h^1_{\text{net}}(p) - \rho q\right), \\
\dot{p} &= \phi\left(h^2_{\text{net}}(q) - \rho p\right).
\end{aligned} \tag{52}$$

where $h^1_{\text{net}}$ and $h^2_{\text{net}}$ are neural networks with $d$-dimensional output and $k$-dimensional output respectively, $\phi$ is a non-linear activation function and $\rho$ is a scalar hyper-parameter.

# C Main Paper Experiments Setting

We select the citation networks Cora(McCallum et al., 2004), Citeseer(Sen et al., 2008), and Pubmed(Namata et al., 2012), and the low-hyperbolicity (Chami et al., 2019) Disease, Airport as the benchmark datasets. The citation dataset is widely used in graph representation learning tasks. We randomly generate 20 samples per class for training, 500 samples for validation, and 1000 samples for testing in each dataset as the same setting in Kipf & Welling (2017a). The low-hyperbolicity datasets Disease and Airport are proposed in Chami et al. (2019), where the Euclidean GNN models cannot learn the node embeddings effectively. We follow the same data splitting and pre-processing in Chami et al. (2019) for Disease and Airport datasets. For the link prediction tasks, we report the best results from different versions of HamGNN. HamGNN (19) for the Disease dataset and (21) for other datasets. This is because we get asynchronous CUDA kernel errors on Disease on our A5000 GPU in this rebuttal period if using (21). We will update all to (21) in the final version after this bug is resolved.

We adjust the model parameters in HamGNN based on the results from the validation data. We use the ADAM optimizer (Kingma & Ba, 2014) with the weight decay as 0.001. We set the learning rate as 0.01 for citation networks and 0.001 for Disease and Airport datasets. The results presented in Table 1 are under the 3 layers HamGNN setting. We report the results by running the experiments over 10 times with different initial random seeds.

HamGNN first compresses the dimension of input features to the fixed hidden dimension (e.g. 64) through a fully connected (FC) layer. Then the obtained hidden features are input to the stacked $H_{\text{net}}$ ODE layers and aggregation layers. The $q$ in Hamiltonian flow is initialized by the node embeddings after the FC layer. **Table 7 shows the implementation details of layer in HamGNN.**

## C.1 ODE solver for Hamiltonian equations

We employ the ODE solver (Chen, 2018) in the implementation of HamGNN. For computation efficiency and performance effectiveness, the fixed-step explicit Euler solver (Chen et al., 2018a) is used in HamGNN. We also compare the influence of ODE solvers and report the results in Table 4. One drawback of the ODE solvers provided in (Chen, 2018) is that they are not guaranteed to have the energy-preserving property in solving the Hamiltonian equations. However, this flaw does not significantly deteriorate our model performance regarding the embedding adaptation to datasets with various structures. Our extensive experiments on the node classification and link prediction tasks

have demonstrated that the solvers provided in (Chen, 2018) are sufficient for our use. We leave the use of Hamiltonian equation solvers for future work to investigate whether solvers with the energy-preserving property can better help graph node embedding or mitigate the over-smoothing problem.

| ODE solver | Euler | Euler | Implicit Adams | Implicit Adams | Dopri5 |
|---|---|---|---|---|---|
| Step size | 0.1 | 0.5 | 0.1 | 0.5 | - |
| Cora | 81.10±1.13 | 81.52±1.27 | 81.62±0.58 | 81.40±0.77 | 81.62±0.58 |

Table 4: Node classification accuracy(%) under different ODE solvers in HamGNN (20).

# D   MORE ABLATION STUDIES AND EXPERIMENTS

## D.1   VANILLA ODE

To demonstrate the advantage of HamGNN's design, we also conduct more experiments that replace the Hamiltonian layer in HamGNN with a vanilla ODE as follows:

$$\dot{q}(t) = \tilde{f}_{\text{net}}(q(t)) \tag{53}$$

where the $\tilde{f}_{\text{net}}(q(t))$ is composed of two FC layers and a non-linear activation function. Compared to the Hamiltonian orbits in Section 4.2, the equation (53) does not include the learnable "momentum" vector for each node and does not follow the Hamiltonian orbits on the cotangent bundle $T^*M$.

## D.2   NODE CLASSIFICATION ON HETEROPHILIC DATASET

| Method | Texas | Wisconsin | Cornell |
|---|---|---|---|
| GCN | $55.56 \pm 3.21$ | $61.96 \pm 1.27$ | $52.35 \pm 7.07$ |
| GAT | $56.22 \pm 6.02$ | $60.36 \pm 5.55$ | $49.61 \pm 6.20$ |
| SAGE | $80.08 \pm 2.96$ | $82.03 \pm 2.77$ | _$81.36 \pm 3.91$_ |
| APPNP | $56.76 \pm 4.58$ | $55.10 \pm 6.23$ | $54.59 \pm 6.13$ |
| GCNII | $61.70 \pm 5.91$ | $62.43 \pm 7.37$ | $52.75 \pm 4.23$ |
| GPRGNN | $72.78 \pm 6.05$ | $69.37 \pm 1.27$ | $76.08 \pm 5.86$ |
| H2GCN | $79.06 \pm 6.36$ | $80.27 \pm 5.41$ | $80.20 \pm 4.51$ |
| GRAND | $75.68 \pm 7.25$ | $79.41 \pm 3.64$ | **$82.16 \pm 7.09$** |
| GraphCON | _$80.00 \pm 3.66$_ | **$84.90 \pm 2.64$** | $75.14 \pm 4.95$ |
| HamGNN | **$81.62 \pm 6.22$** | _$83.92 \pm 4.87$_ | $76.49 \pm 5.10$ |

Table 5: Node classification accuracy(%) on heterophilic datasets. Due to insufficient time during the rebuttal period, the results for GRAND are reported from (Di Giovanni et al., 2022).

To further demonstrate that HamGNN can automatically learn the underlying geometry for datasets with different structures, we include more experiments on the node classification task using heterophilic graph datasets. We select the heterophilic graph datasets Cornell, Texas and Wisconsin from the CMU WebKB [5] project where randomly generated splits of data are provided by Pei et al. (2020). The edges in these graphs represent the hyperlinks between webpages nodes. The labels are manually selected into five classes, student, project, course, staff, and faculty. The features on node are the bag-of-words of the web pages.

For the heterophilic graph datasets, we include the baselines GCN, GAT, SAGE, APPNP (Klicpera et al., 2019), GCNII (Chen et al., 2020b), GPRGNN (Chien et al., 2020), and H2GCN (Zhu et al., 2020a) which are the common baselines for heterophilic graph datasets (Bi et al., 2022). Additionally, we also include GraphCON (Rusch et al., 2022) and GraphCON (Chamberlain et al., 2021b), for

---

[5]http://www.cs.cmu.edu/afs/cs.cmu.edu/project/theo-11/www/wwkb/

| Dataset | Models | 3 layers | 5 layers | 10 layers | 20 layers |
|---------|--------|----------|----------|-----------|-----------|
| Cora | GCN | 80.29±2.29 | 69.87±1.12 | 26.50±4.68 | 23.97±5.42 |
| | HGCN | 78.70±0.96 | 38.13±6.20 | 31.90±0.00 | 26.23±9.87 |
| | HamGNN (20) | 81.52±1.27 | **81.58±0.73** | 79.00±2.17 | 76.20±0.13 |
| | HamGNN (21) | 81.84±0.88 | 81.08±0.16 | **81.40±0.44** | **80.58±0.30** |
| | HamGNN (21) type 2 | **82.10±0.80** | 81.10±0.10 | 81.06±1.49 | 79.26±1.07 |
| Pubmed | GCN | 77.83±0.77 | 76.00±0.87 | 77.53±1.06 | 56.50±12.79 |
| | HGCN | 76.38±0.81 | 77.20±1.05 | 65.90±9.44 | 42.16±2.54 |
| | HamGNN (20) | 78.18±0.54 | 77.86±1.45 | 77.73±1.15 | 76.13±0.80 |
| | HamGNN (21) | 78.83±0.46 | 78.43±0.25 | 78.50±0.61 | 77.20±0.69 |
| | HamGNN (21) type 2 | **79.03±0.58** | **78.46±0.11** | **78.53±0.31** | **77.50±0.44** |

Table 6: Node classification accuracy(%) when increasing the number of layers on the Pubmed dataset.

comparisons. We report the results by running the experiments over 10 times with different initial random seeds for GraphCON and HamGNN, while for the other baselines on the heterophilic graph datasets, we use the results reported in the paper (Bi et al., 2022) (since we use the same experimental setting) due to the time limitation of the rebuttal period.

## D.3 OVER-SMOOTHING

We continue from Section 5.2 to conduct more experiments on the Cora and Pubmed datasets to demonstrate the resilience of HamGNN against over-smoothing. From Table 6, we observe that if the $H_{\text{net}}$ in (21) is convex, HamGNN can even retain its classification ability better than vanilla $H_{\text{net}}$ in (20). One possible reason has been indicated in Section 4.2.3 since now the Hamiltonian formalism degenerates to a Lagrangian formalism with a possible minimization of the dual energy functional (5). In physics, lower energy in most cases indicates a more stable system equilibrium. Moreover, to further show that with the convex $H_{\text{net}}$ in (21) HamGNN can perform better than the vanilla $H_{\text{net}}$ in (20) against over-smoothing, we now include more choices of convex network $H_{\text{net}}$ with different layer sizes and different activation functions (as along as the layer weights from the second layer in $H_{\text{net}}$ are non-negative, and all activation functions in $H_{\text{net}}$ are convex and non-decreasing). The network details of $H_{\text{net}}$ are given in Table 7. The experiment results are shown in Table 6. We clearly observe that for different convex functions and on different datasets, HamGNN with convex $H_{\text{net}}$ nearly keeps the full node classification ability even though we have stacked 20 Hamiltonian layers.

Fig. 3 shows how the node features evolve over 10 layers. We sample a node from the test set of PubMed dataset and input it to three neural ODE-based GNN models, which are 1) HamGNN, 2) an ODE using a positive-definite linear layer as the ODE function, and 3) an ODE using a negative-definite linear layer as the ODE function. Each of these three GNNs contains 10 layers and we compute the node feature magnitude, which is defined to be the $L_2$-norm of the feature vector, and the node feature phase, which is defined to be the cosine similarity between the output feature at the current layer and the input feature to the first layer. We can observe from Fig. 3 that HamGNN has its learned node features change steadily and slowly, while the node features learned by GIL and GCN change abruptly, especially for the feature phases. The features from two nodes of different classes, learned by GIL and GCN, are converging to each other much faster than HamGNN.

## D.4 STABILITY

It is well known that a Hamiltonian system is an energy-conservative system. Our HamGNN inherits this conservative or stability property. The results are shown in Fig. 4, where the experimental setting is the same as those in Fig. 3. The trends are very obvious, where the feature magnitude (you may think of it as feature energy) learned by HamGNN is conserved over layers, while the feature magnitude learned by the positive-definite explodes after a few layers, and the feature magnitude learned by the negative-definite ODE is very close to zero. Regarding the feature phase, the feature phases from two nodes are steadily and slowly approaching each other when using HamGNN, while

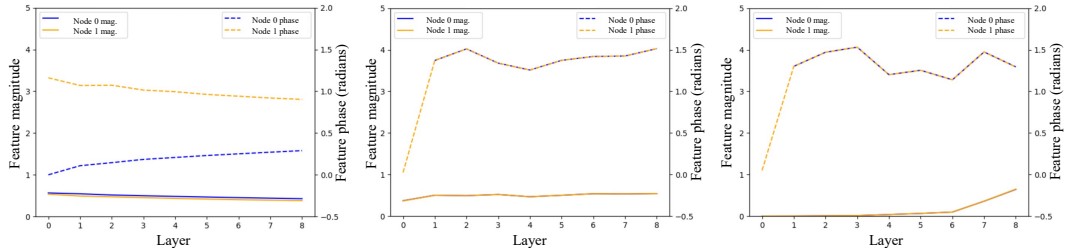

Figure 3: Two nodes from different classes and the evolution of their feature vectors over layers. Left: HamGNN, middle: GIL, right: GCN.

the feature phases learned by the other neural ODEs change abruptly and the difference between two nodes using negative-definite ODE is negligibly small.

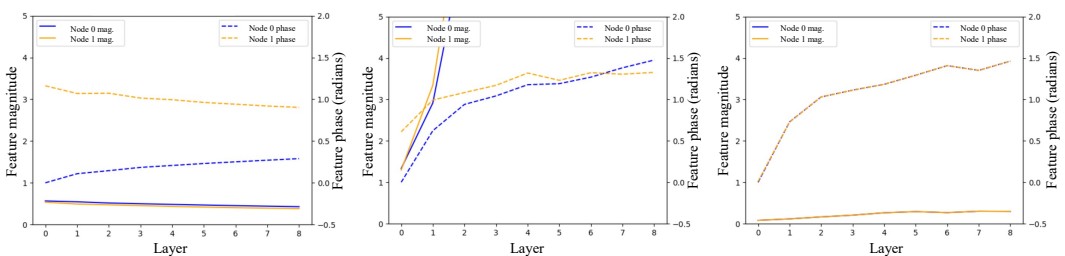

Figure 4: Two nodes from different classes and the evolution of their feature vectors over layers. Left: HamGNN, middle: neural ODE using a positive-definite linear layer, right: neural ODE using a negative-definite linear layer.

Table 7: Neural network modules and the parameters, where $\kappa(x) = 0.5x^2$ if $x > 0$ and $= \exp(x) - 1$ if $x < 0$

| Network | Modules | Activation | Output Channels |
|---|---|---|---|
| (20) | Linear
tanh
Linear | tanh | 1 |
| (21) | ReHU(Amos et al., 2017)
Linear
ReHU
Linear
ReHU
Linear
ReHU | ReHU | 1 |
| (21) type 2 | $\kappa$
Linear
$\kappa$
Linear
$\kappa$
Linear
$\kappa$ | ReHU | 1 |
| (52) | Linear
act1
Linear
act1 | act1 | 576 |
| (22) | Linear
tanh
Linear | tanh | 1 |
| (51) | Linear
tanh
Linear
Sigmoid | tanh
Sigmoid | 64 |
| (23) | Linear
sin
Linear
sin | sin | H:1
W:128 |
| $Q_{\text{net}}$
Raw feature compressing FC | Linear
Linear | identity
identity | 64
64 |
| MLP in Section 5 | Linear
ReLU
Linear
ReLU
Linear | ReLU | Number of classes |

