# OpenReview forum: "NEURAL HAMILTONIAN FLOWS IN GRAPH NEURAL NETWORKS"
_ICLR.cc/2023/Conference — Submitted to ICLR 2023_

### Official Review · Reviewer_XpN8 · 2022-10-22

**Confidence:** 5
**Correctness:** 2
**Technical Novelty And Significance:** 3
**Empirical Novelty And Significance:** 2
**Recommendation:** 3

**Clarity, Quality, Novelty And Reproducibility:**

This paper extends the Hamiltonian network to graph node embedding and classification. The paper writing is not clear. It is unclear in multiple aspects:
1. Unnecessarily abstract: Differential geometry is introduced to reveal that the an quantity is invariant under change of parameterization and thus more fundamental than its representation. This problem, however, is about how to learn optimal representation for the given task. In fact, reading the paper, it is unclear that how the theorems in section 3.2 has helped to develop the Hamiltonian neural network model introduced below. In fact, the entire Hamiltonian neural network can be introduced without mentioning of covector and the vector/covector fields. From the purpose of the paper, i.e. to propose a solution to optimize the node classification task, it is not clear why we add these level of abstraction. The differential geometrical perspective has shown fundamental distinct perspective compared to classical mechanics.

2. Transition from section 2,3  to the proposed framework in section 4 is too sudden. The concept of momentum is a new variable for the network. What is the objective function of the neural network ? it is not clear and the author assume the reader to know this but it is not possible by reading the paper. Also it states “We consider a graph as a discretization of a manifold M” It is unclear why and how. No reference in this part is provided. Is there any tradeoff for this approximation ? it is unclear.

3. The important experiment i.e. the details for resilience to over-smoothing is left in the appendix and only a paper is shown in the main paper. It is not clear as well.

The paper provides a set of codes for the experiment reproduction. The code structure is not very clear and need more comments and documentation for people to use it. From the experiment section of this paper, it is hopeless to reproduce due to lack of too much details in the implementation.


**Strength And Weaknesses:**

The strength of this paper lies in its novel idea of using the Hamiltonian dynamic system in replace of the traditional bipartite graph-based neural network architecture in each layer. By leveraging the energy conservation property of the Hamiltonian, it stabilizes the node embedding process to overcome the over-smoothing issue due to deep architecture of the neural network.

The major weakness of this paper are as below:
1. The paper structure is highly imbalanced. The main contribution of this paper is a novel graph neural architecture in node classification, which is introduced within one paragraph. On the other hands, 3 pages of paper are dedicated to “preliminaries” that neither sufficient nor necessary. It is not sufficient since it requires the readers to finish at least one year worth of study in smooth manifold theory to comprehend all the concepts introduced in the section. It is unnecessary since when it reaches to the implementation of neural network, it comes back to the classical Hamiltonian mechanics which can be introduced without using differential geometry.

Moreover, what is the benefit to view the problem from differential geometrical perspective is not clear to the reader. As stated above, the node embedding itself is certainly not invariant under change of coordination or diffeomorphism, and the goal for this system to be introduced is to learn the stable/optimal node embedding for classification. It is also unclear, after the introduction of concepts on symplectic vs Riemanian, how these two types of manifolds generate different graphs after discretization. The reader is left wondering why we need to introduce a new type of manifold when Riemanian manifold is naturally available.

2. The imbalance of the paper is also reflected in the introduction section. The abstract discussed that the challenge of graph neural network is the oversmoothing and oversquashing. These challenges are not discussed in the introduction section. Instead, the authors discuss differential geometric perspective in node embedding models. It feels that the author is motivated by the geometry, not the problem.

3. The details of the experiment sections are not enough in the main page and many important results are left in the appendix which itself is not clear. For instance, the implementation of Hamiltonian flow part is left with no explanation. The reader is expected to guess what type of ODE solver is involved. It is also not mentioning that normal ODE solver is not guaranteed to have the energy preserving property thus it is not suitable for this experiment. Thus lack of details in implementation makes it impossible to reproduce the result unless to copy the code of the author (which is very uninformative and unclear)


**Summary Of The Paper:**

In this paper, the authors propose to address the oversmoothing and oversquashing problem in graph neural networks by introducing the Hamiltionian flow.

The main contributions are:
1. Introduction of Hamlitonian flow from differential geometry perspective
2. Propose to use Hamiltonian Neural Network in graph node embedding and classification problem
3. By the claim of the paper, the proposed method achieves state-of-the-art performance in node classification task


**Summary Of The Review:**

In summary, i would not recommend this paper for ICLR. There are several issues that the author need to rework:

1. Rephrase the problem and stress the challenge in the introduction section, instead of focussing on the geometry. Introduction is supposed to motivate the reader to follow the paper not to confuse them.

2. Emphasize the intuition over the abstraction. Differential geometry focus on invariance under diffeomorphism and it is important to stress what is invariant under this new formulation.

3. Please push more geometrical concepts into the appendices and swap them with the experiment details. It is the experiments that the readers are more interested in.

4. Providing more details on your implementation which also helps to make abstract concepts concrete. This entire paper is floating in the air now since no concrete formulation is provided.

5. Consider publication in the applied math domains. It is likely that the geometry instead of the graph node classification is the main interest of this paper. It is thus recommended for applied math related domains.

---

> ### Author Response · Authors · 2022-11-19
> **Responses to Clarify Reviewer 4 XpN8's Weakness 1 and 2**
>
> $\textbf{Q1: The paper structure is highly imbalanced. The main contribution of this paper is a novel graph neural architecture in node}$ $\textbf{classification, which is introduced within one paragraph. On the other hands, 3 pages of paper are dedicated to “preliminaries” that}$
> $\textbf{ neither sufficient nor necessary. It is not sufficient since it requires the readers to finish at least one year worth of study in smooth }$
> $\textbf{manifold theory to comprehend all the concepts introduced in the section. It is unnecessary since when it reaches to the }$
> $\textbf{implementation of neural network, it comes back to the classical Hamiltonian mechanics which can be introduced without using}$ $\textbf{differential geometry. }$
>
> $\textbf{Moreover, what is the benefit to view the problem from differential geometrical perspective is not clear to the reader. As stated }$
> $\textbf{above, the node embedding itself is certainly not invariant under change of coordination or diffeomorphism, and the goal for this}$ $\textbf{system to be introduced is to learn the stable/optimal node embedding for classification. It is also unclear, after the introduction of}$
> $\textbf{concepts on symplectic vs Riemanian, how these two types of manifolds generate different graphs after discretization. The reader}$
> $\textbf{is left wondering why we need to introduce a new type of manifold when Riemanian manifold is naturally available.}$
>
>
> **Response 1:**
>
> Thank you very much for this valuable comment.
> We apologize for the imbalanced paper structure in our original submission. As our work is motivated by geometry, we gave sufficient math details in order to make our submission self-contained. Now we realize that too many irrelevant details cause a lot of confusion to readers. Therefore, in the revision, we have removed most of the unnecessary differential geometry concepts in the "Preliminaries" section and thoroughly rewritten this section with much more intuitive motivations. We also thoroughly rewrite the "Proposed Framework" to give more details of the model architecture, implementation, and network setting.
>
> Regarding viewing the problem from a differential geometrical perspective, we are not dealing with the invariant under change of coordination or "symmetry" in physics.
> Thank you for this criticism! Accordingly, we have removed massive differential geometry concepts to make the paper more succinct.
> We now instead provide a clear motivation throughout the revision.
> More specifically, we have added motivation I and motivation II to better explain how the HamGNN relates to hyperbolic GNNs. First, we claim that (cf. Motivation I) hyperbolic exponential map is a solution under fixed metric to the Riemannian geodesic equation and it enjoys a simple closed-form and thus has been widely applied. The geodesic equation is given in equation (8). Furthermore, the geodesic equation can be reformulated as a Hamiltonian function (cf. equation (13) in Motivation II). In other words, the geodesic equation can be considered as a special case of the Hamiltonian equation. Based on this understanding, we can directly learn the embedding manifold using Hamiltonian equations where we treat the Riemannian metric as a learnable diagonal matrix (cf. equation (20) in Section 4.2.1) rather than a fixed constant as assumed in hyperbolic GNNs. As a result of this adaptive metric, HamGNN is capable of learning the potential embedding space of different structured datasets.
>
> We really hope our new writing can make our work more accessible to all the readers including the reviewers!
>
> $\textbf{Q2: The imbalance of the paper is also reflected in the introduction section. The abstract discussed that the challenge of graph}$ $\textbf{neural network is the oversmoothing and oversquashing. These challenges are not discussed in the introduction section. Instead,}$ $\textbf{the authors discuss differential geometric perspective in node embedding models. It feels that the author is motivated by the }$
> $\textbf{geometry, not the problem.}$
>
> **Response 2:**
>
> Sorry for the confusion. We have modified the abstract to emphasize clearly that our main objective is to design a new node embedding strategy that can automatically learn, without extensive tuning, the underlying geometry of any given graph dataset even if it has diverse graph structures.
>
> We are not trying to solve the oversmoothness problem in GNNs. The model resilience to oversmoothing is just a side benefit that has been observed from experiments.
> The purpose is to shed light on this benefit only. Since the oversmoothing problem is not this paper’s main focus, in the revision we do not expand too much on this point and will leave the theoretical analysis for future work.

---

> > ### Author Response · Authors · 2022-11-23
> > **Responses to Clarify Reviewer 4 XpN8's Weakness 3**
> >
> > $\textbf{Q3:The details of the experiment sections are not enough in the main page and many important results are left in the appendix}$ $\textbf{which itself is not clear. For instance, the implementation of Hamiltonian flow part is left with no explanation. The reader is }$ $\textbf{expected to guess what type of ODE solver is involved. It is also not mentioning that normal ODE solver is not guaranteed to have }$
> >  $\textbf{the energy preserving property thus it is not suitable for this experiment. Thus lack of details in implementation makes it impossible}$
> >  $\textbf{to reproduce the result unless to copy the code of the author (which is very uninformative and unclear)}$
> >
> > **Response 3:**
> >
> > Thank you for this feedback. We have included one section to discuss the solver in the appendices Section
> > C.1 ``ODE solver for Hamiltonian equations''. We refer to the review to the appendices for details.
> >
> > We have provided more detailed instructions for the reproduction of our experiments in the paper appendices with all neural layers in Table 7 in the paper.  Furthermore, since we have thoroughly modified the "Proposed Framework" section with a sufficient discussion about the motivation, model architecture, and experimental setups, we hope the reproducibility is now been improved.
> > In the attached Supplementary Material, we also provide sufficient instructions with newly added link prediction experiments and the code for new graph datasets.
> > We hope the details provided in the revision can address the reviewer's concerns.

---

> ### Author Response · Authors · 2022-11-19
> **Responses to Clarify Reviewer 4 XpN8's "Clarity, Quality, Novelty And Reproducibility"**
>
> $\textbf{Q1: Unnecessarily abstract: Differential geometry is introduced to reveal that the an quantity is invariant under change of }$
> $\textbf{parameterization and thus more fundamental than its representation. This problem, however, is about how to learn optimal }$
> $\textbf{representation for the given task. In fact, reading the paper, it is unclear that how the theorems in section 3.2 has helped to develop }$
> $\textbf{the Hamiltonian neural network model introduced below. In fact, the entire Hamiltonian neural network can be introduced without }$
> $\textbf{mentioning of covector and the vector/covector fields. From the purpose of the paper, i.e. to propose a solution to optimize the }$
> $\textbf{node classification task, it is not clear why we add these level of abstraction. The differential geometrical perspective has shown}$
> $\textbf{fundamental distinct perspective compared to classical mechanics.}$
>
> **Response 1:**
> We have addressed the above concerns raised by the reviewer in ``Weaknesses''.
>
> $\textbf{Q2: Transition from section 2,3 to the proposed framework in section 4 is too sudden. The concept of momentum is a new variable }$
> $\textbf{for the network. What is the objective function of the neural network ? it is not clear and the author assume the reader to know this }$
> $\textbf{but it is not possible by reading the paper. Also it states “We consider a graph as a discretization of a manifold M” It is unclear why }$
> $\textbf{and how. No reference in this part is provided. Is there any tradeoff for this approximation ? it is unclear.}$
>
> **Response 2:**
> In the "Motivations and Preliminaries" section in the revision, we have now provided an introduction to Hamiltonian mechanics without mentioning the vector/covector fields where the concept of momentum is well explained.
> We mentioned that a graph is a discretization of a manifold $M$.
> In the revision, the ``discretization'' is now well explained as that we assume the node features are positions on the manifold. We refer the reviewer to Section 4 for the details.
>
> $\textbf{Q3: The important experiment i.e. the details for resilience to over-smoothing is left in the appendix and only a paper is shown in}$ $\textbf{the main paper. It is not clear as well.}$
>
> **Response 3:**
> Thank you for the comments. We do not aim to propose a model that can solve the oversmoothing issue. As a side benefit of our HamGNN, we observe that if more Hamiltonian layers are stacked, HamGNNs still retain the node classification ability, while the classification accuracy of other normal GNNs has dropped a lot. We empirically analyzed this phenomenon in the supplementary.
> Since the oversmoothing problem is not this paper’s main focus, in the revision we do not expand on this point and will leave the theoretical analysis for future work.

---

> ### Author Response · Authors · 2022-11-19
> **Responses to Clarify Reviewer 4 XpN8's "Summary Of The Review"**
>
> **Response:**
> As we have indicated in the summary of the changes we made in the revision, we have done a thorough major revision according to your valuable feedback. We really hope the new writing can make our work more accessible to all the readers, including the reviewers!

---

> ### Author Response · Authors · 2022-11-25
> **Thanks to Reviewer XpN8**
>
> Dear Reviewer XpN8,
>
> Could you please let us know if we could further clarify or provide anything? We believe that we have improved the paper a lot from your previous comments. Please let us know if there is anything we can do to further improve this paper!
>
> Thanks in advance!
>
> Authors

---

> > ### Author Response · Authors · 2022-12-02
> > **Thanks to Reviewer XpN8**
> >
> > Dear Reviewer __XpN8__,
> >
> > Could you please let us know if we could further clarify or provide anything? We believe that we have improved the paper a lot from your previous comments. Please let us know if there is anything we can do to further improve this paper!
> >
> > Thanks in advance!
> >
> > Authors

---

### Official Review · Reviewer_skVS · 2022-10-23

**Confidence:** 4
**Correctness:** 2
**Technical Novelty And Significance:** 2
**Empirical Novelty And Significance:** 1
**Recommendation:** 3

**Clarity, Quality, Novelty And Reproducibility:**

The methodological part of the paper is not clear and not easy to follow.

The authors do rigorously analyze the Hamiltonian flows model, but the idea itself is not novel (please see papers references in my review above).

I am not sure that given the details in the paper I could reproduce the model proposed by the authors.

**Strength And Weaknesses:**

Strengths:

- The authors provide a rigorous theoretical discussion.
- The authors clearly explain why Hamiltonian based GNNs can be useful.

Weaknesses:

- I am not convinced that the proposed method is the first to be based on Hamiltonian (hyperbolic) flows in GNNs. For example:
"PDE-GCN: Novel Architectures for Graph Neural Networks Motivated by Partial Differential Equations"

"Graph-Coupled Oscillator Networks"

also utilize such flows but are not discussed in the paper.

- The experimental section lacks comparison with recent and state-of-the-art methods, which are stronger by a significant margin. For example:
GRAND: Graph Neural Diffusion

Simple and Deep Graph Convolutional Networks

Dirichlet Energy Constrained Learning for Deep Graph Neural Networks

- The experimental section's scope is quite narrow. I think that more experiments on additional datasets (e.g., heterophilic datasets like Cornell and Wisconsin, larger datasets like ogbn-arxiv) are required to measure the benefit of such an approach.

- The authors focus only on transductive datasets (this is stated in text), but it is not clear why. Is it a limitation of the method?

-The paper is hard to follow and in my opinion can be better organized.

- The authors discuss the oversmoothing problem but in the experimental section only report the results with up to 20 layers. I can not draw conclusions based on 20 layers and would recommend the authors to include results with a larger number of layers(e.g, 64 layers).







**Summary Of The Paper:**

The authors propose a Hamiltonian based feature learning in GNNs to overcome the oversmoothing and oversquashing phenomena that are evident in many existing GNNs.
A rigorous theoretical discussion is provided and several experiments are conducted.

**Summary Of The Review:**

The paper sheds more light and analyzes the Hamiltonian flows approach in GNNs. Several experiments are conducted, but they are far from current model accuracy and performance, and proper discussion of existing and relevant model is missing, including a quantitative comparison with recent GNNs.

---

> ### Author Response · Authors · 2022-11-19
> **Responses to Clarify Reviewer 3 skVS's Weakness 1-5**
>
> $\textbf{Q1: I am not convinced that the proposed method is the first to be based on Hamiltonian (hyperbolic) flows in GNNs. For example:}$ $\textbf{"PDE-GCN: Novel Architectures for Graph Neural Networks Motivated by Partial Differential Equations"}$
> $\textbf{"Graph-Coupled Oscillator Networks" also utilize such flows but are not discussed in the paper.}$
>
>  $\textbf{The experimental section lacks comparison with recent and state-of-the-art methods, which are stronger by a significant margin. }$
>  $\textbf{For example: GRAND: Graph Neural Diffusion}$
>
>
> **Response 1:**
> Thank you for your valuable comments! We compare HamGNN with GRAND and GraphCON, and the results are reported in Table 1 in the paper revision. We follow the convention in graph node embedding literature, and **do not** use the largest connected component of datasets that however have been used in GRAND and GraphCON. We also do not use rewire method for a fair comparison. We can see that the HamGNN still has the best node classification results on the hyperbolic datasets Airport and Disease, and has a comparable result on Euclidean datasets Cora, Citeseer, and Pubmed.
>
> Please also note that there is a fundamental difference between HamGNN and graph neural flows such as GRAND, GraphCON and PDE-GCN. The graph neural flows consider the message passing process as a continuous diffusion process so that in their models they wrap message aggregation function (taking node features and adjacency matrix as input), e.g., attention-based aggregation function in GAT, into ODE. However, in our HamGNN, you may think the Hamiltonian ODE as a new node embedding layer (taking only node features as input) which does not aggregate messages among neighbouring nodes. Hence, in our model, we perform node embedding first and then message aggregation, where these two steps are separated.
>
> We refer the reviewer to the "Motivations and Preliminaries" section for our graph node embedding motivations from exponential maps and geodesic equations which may clarify the difference between our model and the physics-inspired Graph neural diffusion works.  We also have provided a detailed comparison between our work and the Graph neural diffusion in the "Related Work" section.
>
>
>  $\textbf{Q2: Simple and Deep Graph Convolutional Networks. Dirichlet Energy Constrained Learning for Deep Graph Neural Networks}$
>
> **Response 2:**
> Thank you for your valuable comments!  The models in these two papers are designed to solve the oversmoothing issue in GNNs. Although we have also included oversmoothing results, this is not this paper's main contribution. We now made our contribution much clear in the revision. To summarize, our main contribution is to propose a node embedding strategy that can automatically learn, without extensive tuning, the underlying geometry of any given graph dataset even if it has diverse geometry.
>
>  $\textbf{Q3: The experimental section's scope is quite narrow. I think that more experiments on additional datasets (e.g., heterophilic}$  $\textbf{datasets like Cornell and Wisconsin, larger datasets like ogbn-arxiv) are required to measure the benefit of such an approach.}$
>
> **Response 3:**
>  Thank you for your valuable comments!  We include the results of the heterophilic datasets in Table R1  in the responses to Reviewer FBem, we re-run the GraphCon code by using the hyperparameter in https://github.com/tk-rusch/GraphCON/blob/main/src/heterophilic_graphs/best_params.py. We observe that HamGNN still has competitive results compared with SOTA on the heterophilic datasets. We also include the link prediction results in Table R2 in the responses.
>  We also would like to emphasize that the two standard graph node embedding downstream tasks, node classification and link prediction, have been widely used for the performance validation of graph node embedding GNNs like GIL, HGNN, HGCN and HGAT.
>
>  $\textbf{Q4: The authors focus only on transductive datasets (this is stated in text), but it is not clear why. Is it a limitation of the method?}$
>
> **Response 4:**
> Thank you for your valuable comments! It is not a limitation of HamGNN, our model can also be adapted for inductive learning or graph classification. We only consider transductive learning in the paper because we followed the same experiment setting in graph node embedding works like GIL, HGNN, HGCN and HGAT.
>
>  $\textbf{Q5: The paper is hard to follow and in my opinion can be better organized.}$
>
> **Response 5:**
> Thank you for your valuable comments!
> We have revised the paper structure. We refer the reviewer to our new manuscript which has been uploaded to OpenReview.

---

> > ### Author Response · Authors · 2022-11-23
> > **Responses to Clarify Reviewer 3 skVS's Weakness 6**
> >
> > $\textbf{Q6: About oversmoothing problem but in the experimental section only report the results with up to 20 layers.}$
> >   $\textbf{I can not draw conclusions based on 20 layers and would recommend the authors to include results with a larger}$
> > $\textbf{ number of layers(e.g, 64 layers).}$
> >
> > **Response 6:**
> > We do not aim to propose a model that can solve the oversmoothing issue. As a side benefit of our HamGNN, we only observe that if more Hamiltonian layers are stacked, HamGNNs still retain the node classification ability, while the classification accuracy of other normal GNNs has dropped a lot.
> > We empirically analyzed this phenomenon in the supplementary.
> > However since the oversmoothing problem is not this paper’s main focus, in the revision, we do not expand on this point and will leave the theoretical analysis for future work.

---

> > > ### Author Response · Authors · 2022-11-27
> > > **more experiments for resiliency to over-smoothing**
> > >
> > > Dear Reviewer skVS,
> > >
> > > We have done experiments with up to 64 layers. We refer you to our response to Reviewer FBem for the results and our explanation. Note, our HamGNN __does not have a skip connection__ and we do not use any tricks like edge-dropping.
> > >
> > > Best,
> > >
> > > Authors

---

> ### Author Response · Authors · 2022-11-19
> **Responses to Clarify Reviewer 3 skVS's "Clarity, Quality, Novelty And Reproducibility"**
>
>  $\textbf{Q: The methodological part of the paper is not clear and not easy to follow.}$
>
>  $\textbf{The authors do rigorously analyze the Hamiltonian flows model, but the idea itself is not novel}$
>   $\textbf{(please see papers references in my review above).}$
>
>  $\textbf{I am not sure that given the details in the paper I could reproduce the model proposed by the authors.}$
>
> **Response:**
> Thank you for your feedback. We have provided more detailed instructions for the reproduction of our experiments in the paper appendices with all neural layers in Table 7 in the paper. Furthermore, since we have thoroughly modified the "Proposed Framework" section with a sufficient discussion about the motivation, model architecture, and experimental setups, we hope the reproducibility is now been improved.
> In the attached Supplementary Material, we also provide sufficient instructions with newly added link prediction experiments and the code for new graph datasets.
> We hope the details provided in the revision can address the reviewer's concerns.

---

> ### Author Response · Authors · 2022-11-19
> **Table R2 of New Experimental Results**
>
> | Method |        Disease         |        Airport         |         Pubmed         |        Citeseer        |          Cora          |     |     |
> |:------:|:----------------------:|:----------------------:|:----------------------:|:----------------------:|:----------------------:|:---:|:---:|
> |  MLP   |    83.37 $\pm$ 5.04    |    87.04 $\pm$ 0.56    |    88.69 $\pm$ 1.59    |    89.65 $\pm$ 1.00    |    91.07 $\pm$ 0.56    |     |     |
> |  HNN   |    81.37 $\pm$ 8.78    |    86.06 $\pm$ 2.08    |    94.69 $\pm$ 0.25    |    89.83 $\pm$ 0.39    |    92.83 $\pm$ 0.76    |     |     |
> |  GCN   |    60.38 $\pm$ 2.51    |    90.97 $\pm$ 0.65    |    91.37 $\pm$ 0.09    |    93.20 $\pm$ 0.28    |    92.89 $\pm$ 0.77    |     |     |
> |  GAT   |    62.03 $\pm$ 1.58    |    91.05 $\pm$ 0.83    |    91.03 $\pm$ 0.67    |    93.83 $\pm$ 0.65    |    93.34 $\pm$ 0.50    |     |     |
> |  SAGE  |    68.02 $\pm$ 0.43    |    91.40 $\pm$ 0.88    |    93.61 $\pm$ 0.26    |    93.37 $\pm$ 0.88    |    92.94 $\pm$ 0.40    |     |     |
> |  SGC   |    59.83 $\pm$ 4.01    |    89.72 $\pm$ 0.82    |    92.16 $\pm$ 0.13    |    94.78 $\pm$ 0.77    |    93.15 $\pm$ 0.22    |     |     |
> |  HGNN  |    60.20 $\pm$ 1.14    |    92.46 $\pm$ 0.20    |    93.09 $\pm$ 0.09    |    90.35 $\pm$ 0.57    |    92.05 $\pm$ 0.33    |     |     |
> |  HGCN  |    78.09 $\pm$ 2.79    |    94.28 $\pm$ 0.20    | ***96.79 $\pm$ 0.01*** |    93.60 $\pm$ 0.14    |    94.10 $\pm$ 0.05    |     |     |
> |  HGAT  |    76.32 $\pm$ 3.41    |    94.64 $\pm$ 0.51    |  **96.86 $\pm$ 0.03**  |    93.45 $\pm$ 0.25    |    94.96 $\pm$ 0.36    |     |     |
> |  GIL   |  **99.97 $\pm$ 0.08**  | ***97.92 $\pm$ 2.64*** |    91.22 $\pm$ 3.25    | ***95.99 $\pm$ 8.89*** | ***97.78 $\pm$ 2.31*** |     |     |
> | HamGNN | ***99.73 $\pm$ 0.26*** |  **99.99 $\pm$ 0.01**  |    92.15 $\pm$ 0.30    |  **99.99 $\pm$ 0.00**  |  **98.20 $\pm$ 1.73**  |     |     |
>
> Table R2: Link prediction ROC(\%). The best and the second-best result are highlighted in **bold** and ***bold and italic***, respectively.
>
> **Notes:** Due to insufficient time during the rebuttal period, we are not able to provide link prediction results for other GNNs that do not have the open-source link prediction code. We however have included **all** the GNNs with open-source link prediction code and will update the other GNNs' code to perform the link prediction task in the future.

---

### Official Review · Reviewer_FBem · 2022-10-24

**Confidence:** 3
**Correctness:** 3
**Technical Novelty And Significance:** 2
**Empirical Novelty And Significance:** 2
**Recommendation:** 5

**Clarity, Quality, Novelty And Reproducibility:**

The authors demonstrate a deep understanding of the subject matter and present an interesting framework for graph neural networks. The proposed method seems to be a novel application of Hamiltonian flows to graph neural networks and the novelty of the work is positioned well by the related works section.

That being said, I find the paper difficult to follow, and several typos can be found in the text. Furthermore, the mathematical theory used is not sufficiently motivated and the advantage of the method is not sufficiently clarified — especially given that the performance of this method is not better than the current state-of-the-art methods (see the previous section).

The paper spends a lot of space presenting definitions and theorems from differential geometry and Hamiltonian mechanics using notation that may be greatly simplified and clarified. For example, the definition of $q$ on page 3 is confusing, and the definition immediately following it is unclear. Also, the fact that definitions have no number or name associated makes it difficult to refer to them.

Very little space is dedicated to discussing the actual method being proposed, and generally, I find that it is difficult to locate the salient information in the text. The notational conventions used may not be clear to members of the community, e.g., the coproduct or $U \rightarrow q(U)$. It is also recommended to include a citation of the respective papers in comparison tables since model names tend to be very similar to each other and can be confusing.

The authors have provided code to facilitate reproducibility. The proposed architectures, experimental setups, and training parameters are not listed fully in the main paper and appendices. Perhaps the supplementary materials are sufficient to reproduce this research. However, I did not attempt to run the code.


**Strength And Weaknesses:**

The authors propose an interesting application of Hamiltonian flows and differential geometry to graph neural networks. However, most of the models they compare with in the experiments section are not the newest and despite the paper’s claims, no longer represent the state of the art. I recommend the authors compare to more modern graph neural networks including GCNII (2020), LGCN (which is cited in the related works section but not compared to), GRAND (Chamberlain et al.), PDE-GCN (Eliasof et al. 2021), GraphCon (Rusch et al. 2021), EGNN (Zhou et al 2021) or newer. According to the results listed, the proposed method does not achieve SOTA performance compared to the works cited above.

In the absence of SOTA performance, the paper does not otherwise substantiate the use of the mathematical theory. The contribution of the proposed methods is not discussed in terms of any other metric, such as reduced running times or the number of parameters, etc. Furthermore, the authors explore different network depths, but the point of obtaining deeper networks is lost when accuracy is not comparable to other models of similar depth.

The paper is difficult to follow, and the method of Hamiltonian flows is not clearly motivated for the task of node classification. See the next section for details.


**Summary Of The Paper:**

The authors present a novel application of differential geometry and Hamiltonian flows to graph neural networks and demonstrate its applicability to the common GNN problem of node classification. The theory gives rise to a GNN model which is then shown to perform relatively well on standard datasets and somewhat to resist over-smoothing as the depth of the network increases.


**Summary Of The Review:**

I find the paper very interesting, however the authors do not present compelling motivation or outstanding experimental results for this reviewer to consider using the methods. They do not discuss the methods and experimental setups in sufficient detail, and do not provide comparisons to state-of-the-art methods.

The paper is well presented, though could benefit from more editing work to fix typos, clarify notations, and add necessary details to motivate its inclusion into scientific archives.

---

> ### Author Response · Authors · 2022-11-19
> **Responses to Clarify Reviewer 2 FBem's Weakness**
>
> $\textbf{Q1: The authors propose an interesting application of Hamiltonian flows and differential geometry to graph neural networks. However,}$
> $\textbf{most of the models they compare with in the experiments section are not the newest and despite the paper’s claims, no longer}$ $\textbf{represent the state of the art. I recommend the authors compare to more modern graph neural networks including GCNII (2020),}$ $\textbf{LGCN (which is cited in the related works section but not compared to), GRAND (Chamberlain et al.), PDE-GCN (Eliasof et al. 2021),}$
> $\textbf{GraphCon (Rusch et al. 2021), EGNN (Zhou et al 2021) or newer. According to the results listed, the proposed method does not }$
> $\textbf{achieve SOTA performance compared to the works cited above.}$
>
> **Response 1:** Thank you for your comment. We have added new experimental results in Table R1(this table also has been included as Table 5 in the paper revision)  in the responses to Reviewer FBem and Table 1 in the paper revision where new models are compared with ours (due to insufficient time during the rebuttal period, we are not able to compare to all of the models mentioned by the reviewer). Please also note the difference between our HamGNN and graph neural flows such as GRAND, PDE-GCN, and GraphCON. In these methods, they wrap the message passing function, e.g., constant aggregation function in GCN and attention-based aggregation function in GAT, into an ODE function. In contrast, HamGNN separates the node embedding process and node aggregation process. That is, we use the Hamiltonian ODE function only for embedding nodes which is then followed by a separate node aggregation step.
> We refer the reviewer to the "Motivations and Preliminaries" section for our graph node embedding motivations from exponential maps and geodesic equations. We also have provided a detailed comparison between our work and the physics-inspired Graph Neural diffusion in the "Related Work" section.
>
> $\textbf{Q2: In the absence of SOTA performance, the paper does not otherwise substantiate the use of the mathematical theory. The}$ $\textbf{contribution of the proposed methods is not discussed in terms of any other metric, such as reduced running times or the number}$
> $\textbf{of parameters, etc. Furthermore, the authors explore different network depths, but the point of obtaining deeper networks is lost}$ $\textbf{when accuracy is not comparable to other models of similar depth.}$
>
> **Response 2:** Thank you for your criticism.
> In the  Abstract of the revision, we now emphasize clearly that our goal is to improve the performance of downstream graph learning tasks by enabling an adaptive node embedding mechanism that is expected to suit different structures of graph data. To make our contribution more clear, in our revision, we have highlighted two main motivations, provided more details about how our model is related to the existing graph node embedding GNNs, and explained what are the advantages of our model as compared to the existing GNNs, especially the hyperbolic GNNs. We also extend our Experiments section in our revision where more tasks and datasets are included with more explanation.
>
> $\textbf{Q3: The paper is difficult to follow, and the method of Hamiltonian flows is not clearly motivated for the task of node classification.}$
> $\textbf{ See the next section for details.}$
>
> **Response 3:** Thank you for your comments. We refer the reviewer to the responses to "Clarity, Quality, Novelty And Reproducibility".

---

> > ### Comment · Reviewer_FBem · 2022-11-25
> > **Discussion re added results**
> >
> > Dear Authors,
> >
> > Thank you for adding the results, but I find your argument, "due to insufficient time during the rebuttal period, we are not able to compare to all of the models mentioned by the reviewer" a bit unclear. What I mean by "compare" is to copy the accuracies from the corresponding papers into your tables, and if a result is missing in the original paper, you may place something like a dash sign "-" to indicate it's missing. Looking at the original papers and copying numbers does not take time. You need not perform all the other experiments from scratch, and furthermore, getting the right hyperparameters to get those results is nearly impossible for so many methods. Moreover, it is customary to cite relevant papers if being pointed to.
> >
> > What's more disturbing is the inconsistency between the results presented in your paper and the actual results provided by the authors. For example: GPRGNN in https://arxiv.org/pdf/2006.07988.pdf.
> > Original results for Texas | Wisconsin | Cornell:
> >
> > Original:         92.92±0.61 | - | 91.36±0.70
> >
> > Your results:  72.78±6.05	| 69.37±1.27	| 76.08±5.86
> >
> > BTW - the reported variances are very high compared to GPRGNN.
> >
> > Similarly with GCNII (https://arxiv.org/pdf/2007.02133.pdf):
> >
> > original:                       77.84 (32)  | 81.57 (16)  | 76.49 (16)
> >
> > your reported results: 61.70±5.91 |	62.43±7.37  | 52.75±4.23
> >
> > It seems that the authors reran many experiments and did not achieve the originally reported accuracies. If compared to the original accuracies reported in the original papers, the proposed method is not competitive. I cannot change my score in light of this. Please revise your results, or explain.
> >
> > Another small remark: when other papers test the resiliency to over-smoothing, they usually use up to 64 layers. The authors show up to 20, and a slight degradation in the accuracy is observed. I am not sure that the method is indeed resilient to over-smoothing. What happens with 64 layers? This is a minor comment compared to the previous one.

---

> > > ### Author Response · Authors · 2022-11-25
> > > **The "uncompetitive" phenomenon is mainly due to different data split settings.**
> > >
> > > Thank you for your response! We are glad we got some responses from you and are willing to address them!
> > >
> > > __Response 1.__  After tedious investigations, we first would like to point out that for heterophilic datasets, there are two types of data splitting in the literature:
> > >
> > > >(a). multiple 60\%/20\%/20\% random splits (GPRGNN [6] uses this setting)
> > > >
> > > >(b). 10 fixed 48\%/32\%/20\% splits (we use this setting)
> > >
> > > Data splitting (a) has been used in the papers like [6][7], while splitting (b) has been used in the papers like [2][3][4][5][7]. However, in the literature, the above splittings are not that clear since the pioneering work [2] claims that the ratios are 60\%/20\%/20\%, which is different from the actual data splits shared on GitHub (See the clarification footnotes in [3] page 6 and in [4] page 8).
> > >
> > > The difference is clearly presented in work [7] where it presents two tables: Table 2 shows the models' performance under data splitting (a) while Table 3 shows the models' performance under data splitting (b).
> > > The ``92.92±0.61 | - | 91.36±0.70`` of GPRGNN for "Texas | Wisconsin | Cornell" pointed out by you is from [6] under data splitting (a) and has been shown in [7] Table 2. While in [7] Table 3, under setting (b), GPRGNN shows the accuracy of `` 81.35 ± 5.32  | 82.55 ± 6.23 | 78.11 ± 6.55``. Note, our HamGNN get ``81.62  ±  6.22 | 83.92 ± 4.87  |  76.49± 5.10 ``.
> > > This also has been presented in our new Table R1-Corrected.
> > >
> > > From our new Table R1-Corrected, we still observe that our method is **competitive** for GNNs that are designed specifically for heterophilic datasets.
> > >
> > > ***Overall, we conclude our model achieves SOTA on the hyperbolic datasets: Disease and Airport; and gets competitive performance (among the top5, see our Table 1 in the paper revision and Table R1-Corrected in the rebuttal ) on the homophilic and heterophilic datasets: Pubmed, Citeseer, Cora, Cornell, Texas and Wisconsin.***
> > >
> > > Finally, we would still like to __``emphasize``__ that "we do not aim to outperform all the baselines or other general-purpose GNN models on specific datasets. Instead, our objective is to design a new node embedding strategy that can **automatically learn**, without extensive tuning, the underlying geometry of any given graph dataset even if it has **diverse geometries**." (See our paper revision.)
> > > We promise to also update Table 5 in the paper revision to Table R1-Corrected.
> > >
> > >
> > > __Response 2.__  As mentioned in our revision on page 20, "for the other baselines on the heterophilic graph datasets, we use the results reported in the paper [1] (Bi et al., 2022)", we used the results reported in [1] (Bi et al., 2022) for all the baselines (except GraphCON) in Table R1. In the past few hours, we have also contacted the main author of this paper, and he confirmed that there exists some difference between the reported results and rerun results for some methods (such as GPRGNN and GCNII) even after performing hyperparameter optimization. He claims that such inconsistent results may come from the fact that they [1] use Pytorch implementation whereas original papers use Tensorflow implementation.
> > >
> > > Despite the potential reproducible issues for some baseline methods, we decide to discard Table R1 and compare the results of the baselines that have been widely and consistently reported in many works like [2][3][4][5][7] in Table R1-corrected.
> > >
> > > Even now, our HamGNN is not the best,  we do observe that, without extensive tuning, it has automatically learned the geometries possessed by heterophilic datasets that are essentially different from the homophilic and low hyperbolicity datasets reported in our paper revision Table 1.
> > >
> > >
> > >
> > >
> > >
> > >
> > >
> > >
> > >
> > >
> > >
> > > [1] Bi, Wendong, et al. "Make Heterophily Graphs Better Fit GNN: A Graph Rewiring Approach." arXiv preprint arXiv:2209.08264 (2022). https://arxiv.org/pdf/2209.08264.pdf
> > >
> > > [2] Pei, Hongbin, et al. "Geom-gcn: Geometric graph convolutional networks." arXiv preprint arXiv:2002.05287 (2020). https://arxiv.org/pdf/2002.05287.pdf
> > >
> > > [3] Yan, Yujun, et al. "Two sides of the same coin: Heterophily and oversmoothing in graph convolutional neural networks." arXiv preprint arXiv:2102.06462 (2021). https://arxiv.org/pdf/2102.06462.pdf
> > >
> > > [4] Zhu, Jiong, et al. "Beyond homophily in graph neural networks: Current limitations and effective designs." Advances in Neural Information Processing Systems 33 (2020): 7793-7804.
> > >
> > > [5] Rusch, T. Konstantin, et al. "Graph-coupled oscillator networks." International Conference on Machine Learning. PMLR, 2022. https://proceedings.mlr.press/v162/rusch22a/rusch22a.pdf
> > >
> > >
> > > [6] (GPRGNN) Chien, Eli, et al. "Adaptive universal generalized pagerank graph neural network." arXiv preprint arXiv:2006.07988 (2020). https://arxiv.org/pdf/2006.07988.pdf
> > >
> > > [7]  Luan, Sitao, et al. "Revisiting Heterophily For Graph Neural Networks." arXiv preprint arXiv:2210.07606 (2022). https://arxiv.org/pdf/2210.07606.pdf

---

> > > ### Author Response · Authors · 2022-11-25
> > > **experiments for resiliency to over-smoothing**
> > >
> > > | Dataset |   Models    |     3 layers     |     5 layers     |    10 layers     |    20 layers     |    32 layers     |    64 layers     |     |
> > > |:--------|:-----------:|:----------------:|:----------------:|:----------------:|:----------------:|:----------------:|:----------------:|:---:|
> > > | Cora    | HamGNN (21) | 81.84 $\pm$ 0.88 | 81.08 $\pm$ 0.16 | 81.40 $\pm$ 0.44 | 80.58 $\pm$ 0.30 | 75.35 $\pm$ 0.43 | 71.52 $\pm$ 1.20 |     |
> > >
> > > __Table R3:__ Node classification accuracy(\%) when increasing the number of layers on the Cora dataset (Extended experiments from Table 3 in the paper revision). Code for oversmoothness experiments with 32 and 64 layers is provided in the anonymous github https://github.com/hamgnn/hamgnn.
> > >
> > >
> > > **oversmoothing in the literature:** To prevent the oversmoothing issue in GNN, EGNN [1] proposes the weight controlling, residual connection, and trainable shift ReLU activation function in GCN. The weight of the EGCN layer is initialized with an identity matrix, and also the regularization loss is added to penalize the distances between the trainable weights and initialized weights. EGNN adopts residual connections to the initial layer and the previous layer. GCNII [2] analyzes the oversmoothing problem from the spectral perspective, but it still applies an initial residual that constructs a skip connection from the input layer, and the identity mapping that adds an identity matrix to the weight matrix at each layer.
> > > DropEdge [3] suggests that by randomly removing out a few edges from the input graph, one can relieve the impact of over-smoothing.
> > >
> > > **our HamGNN:** In contrast to the above works, HamGNN **does not** apply these training methods that are designed for deep layers. We try to demonstrate the natural benefits of HamGNN to mitigate the oversmoothing issue and explain this in Section D.3 of the paper revision.
> > >   We have included the over-smoothing experiments with 64-layer HamGNN in Table R3 where we can clearly observe the resiliency to over-smoothing.
> > >  Please especially note that HamGNN **does not** have a skip connection between layers which has been widely adopted in [1][2] to defend against oversmoothness.
> > > In our work, in each layer, we use the Hamiltonian equation to learn a suitable node embedding space.
> > > *We therefore conclude that the oversmoothing problem of GNNs can be mitigated if the node features evolve through Hamiltonian orbits in our node embedding layer. This phenomenon is essentially different from the above works [1][2][3].*
> > >
> > >
> > >
> > >
> > >
> > >
> > > [1] Zhou, Kaixiong, et al. "Dirichlet energy constrained learning for deep graph neural networks." Advances in Neural Information Processing Systems 34 (2021): 21834-21846.
> > >
> > > [2] Chen, Ming, et al. "Simple and deep graph convolutional networks." International Conference on Machine Learning. PMLR, 2020.
> > >
> > > [3] Rong, Yu, et al. "Dropedge: Towards deep graph convolutional networks on node classification." arXiv preprint arXiv:1907.10903 (2019).

---

> > > ### Author Response · Authors · 2022-11-26
> > > **Dear Reviewer FBem, we have finished the new responses to your questions**
> > >
> > > Dear Reviewer FBem, we have finished the new responses to your questions. Thank you for your responses again! We are looking forward to discussing any further questions or suggestions!
> > >
> > > Thank you sincerely for your time and effort in reviewing our paper!

---

> > > > ### Author Response · Authors · 2022-12-02
> > > > **Discussion with Reviewer FBem**
> > > >
> > > > Dear Reviewer __FBem__,
> > > >
> > > > We deeply appreciate your comments that have been used to improve our work. We would like to know whether our responses have addressed your new concerns. We are also happy to answer if you have any further questions.
> > > >
> > > > Thank you!
> > > >
> > > > Best,
> > > >
> > > > Paper5782 Authors

---

> > > > > ### Author Response · Authors · 2022-12-08
> > > > > **regarding new responses to your new questions**
> > > > >
> > > > > Dear Reviewer __FBem__,
> > > > >
> > > > > We deeply appreciate your comments that have been used to improve our work. We would like to know whether our responses have addressed your new concerns. We are also happy to answer if you have any further questions.
> > > > >
> > > > > Thank you!
> > > > >
> > > > > Best,
> > > > >
> > > > > Paper5782 Authors

---

> ### Author Response · Authors · 2022-11-19
> **Responses to Clarify Reviewer 2 FBem's "Clarity, Quality, Novelty And Reproducibility"**
>
> $\textbf{Q1: That being said, I find the paper difficult to follow, and several typos can be found in the text. Furthermore, the mathematical}$ $\textbf{theory used is not sufficiently motivated and the advantage of the method is not sufficiently clarified — especially given that the}$ $\textbf{performance of this method is not better than the current state-of-the-art methods (see the previous section).}$
>
> $\textbf{The paper spends a lot of space presenting definitions and theorems from differential geometry and Hamiltonian mechanics using}$
> $\textbf{notation that may be greatly simplified and clarified. For example, the definition of  on page 3 is confusing, and the definition}$ $\textbf{immediately following it is unclear. Also, the fact that definitions have no number or name associated makes it difficult to refer to them.}$
>
> **Response 1:** Thank you for your criticism. We apologize for the poor exposition in our original submission which causes a lot of confusion. To facilitate exposition, now we have rewritten the manuscript where we emphasized the motivations in a more intuitive manner at the beginning, significantly condensed the introduction of differential geometry, and provided more details about the model architecture.
>
> $\textbf{Q2: Very little space is dedicated to discussing the actual method being proposed, and generally, I find that it is difficult to locate the}$ $\textbf{salient information in the text. The notational conventions used may not be clear to members of the community, e.g., the coproduct }$
> $\textbf{or $U\rightarrow q({U})$. It is also recommended to include a citation of the respective papers in comparison tables since}$ $\textbf{model names tend to be very similar to each other and can be confusing.}$
>
> **Response 2:** Thank you for your criticism. In our revision, we have provided much more details about the model architecture and added references to the methods which we compared in the tables.
>
> $\textbf{Q3: The authors have provided code to facilitate reproducibility. The proposed architectures, experimental setups, and training}$ $\textbf{parameters are not listed fully in the main paper and appendices. Perhaps the supplementary materials are sufficient to reproduce}$
> $\textbf{ this research. However, I did not attempt to run the code.}$
>
> **Response 3:**
> Thank you for your feedback. We have provided more detailed instructions for the reproduction of our experiments in the paper appendices with all neural layers in Table 7 in the paper. Furthermore, since we have thoroughly modified the "Proposed Framework" section with a sufficient discussion about the motivation, model architecture, and experimental setups, we hope the reproducibility is now been improved.
> In the attached Supplementary Material, we also provide sufficient instructions with newly added link prediction experiments and the code for new graph datasets.
> We hope the details provided in the revision can address the reviewer's concerns.

---

> ### Author Response · Authors · 2022-11-19
> **Responses to Clarify Reviewer 2 FBem's "Summary Of The Review"**
>
> $\textbf{Q: I find the paper very interesting, however the authors do not present compelling motivation or outstanding experimental results for}$
> $\textbf{ this reviewer to consider using the methods. They do not discuss the methods and experimental setups in sufficient detail, and do}$
> $\textbf{ not provide comparisons to state-of-the-art methods.}$
> $\textbf{The paper is well presented, though could benefit from more editing work to fix typos, clarify notations, and add necessary details}$
> $\textbf{to motivate its inclusion into scientific archives.}$
>
> **Response:** Thank you very much for your criticism. In our revision, we provide sufficient discussion about the motivation, model architecture, and experimental setups while significantly condensing unnecessary differential geometry definitions.

---

> ### Author Response · Authors · 2022-11-19
> **Table R1 of New Experimental Results**
>
>
> |  Method  |         Texas          |       Wisconsin        |        Cornell         |     |
> |:--------:|:----------------------:|:----------------------:|:----------------------:|:---:|
> |   GCN    |    55.56 $\pm$ 3.21    |    61.96 $\pm$ 1.27    |    52.35 $\pm$ 7.07    |     |
> |   GAT    |    56.22 $\pm$ 6.02    |    60.36 $\pm$ 5.55    |    49.61 $\pm$ 6.20    |     |
> |   SAGE   |    80.08 $\pm$ 2.96    |    82.03 $\pm$ 2.77    | ***81.36 $\pm$ 3.91*** |     |
> |  APPNP   |    56.76 $\pm$ 4.58    |    55.10 $\pm$ 6.23    |    54.59 $\pm$ 6.13    |     |
> |  GCNII   |    61.70 $\pm$ 5.91    |    62.43 $\pm$ 7.37    |    52.75 $\pm$ 4.23    |     |
> |  GPRGNN  |    72.78 $\pm$ 6.05    |    69.37 $\pm$ 1.27    |    76.08 $\pm$ 5.86    |     |
> |  H2GCN   |    79.06 $\pm$ 6.36    |    80.27 $\pm$ 5.41    |    80.20 $\pm$ 4.51    |     |
> |  GRAND   |    75.68 $\pm$ 7.25    |    79.41 $\pm$ 3.64    |  **82.16 $\pm$ 7.09**  |     |
> | GraphCON | ***80.00 $\pm$ 3.66*** |  **84.90 $\pm$ 2.64**  |    75.14 $\pm$ 4.95    |     |
> |  HamGNN  |  **81.62 $\pm$ 6.22**  | ***83.92 $\pm$ 4.87*** |    76.49 $\pm$ 5.10    |     |
>
> **Table R1:** Node classification accuracy(%) on *heterophilic datasets*. The best and
> the second-best results are highlighted in **bold** and ***bold
> and italic***, respectively.

---

> > ### Author Response · Authors · 2022-11-25
> > **Table R1-Corrected**
> >
> > |  Method  |   Cornell    |  Wisconsin   |    Texas     |     |
> > |:--------:|:------------:|:------------:|:------------:|:---:|
> > | Geom-GCN | 60.54 ± 3.67 | 64.51 ± 3.66 | 66.76 ± 2.72 |     |
> > |  H2GCN   | 82.70 ± 5.28 | 87.65 ± 4.98 | 84.86 ± 7.23 |     |
> > |  GPRGCN  | 78.11 ± 6.55 | 82.55 ± 6.23 | 81.35 ± 5.32 |     |
> > |  FAGCN   | 76.76 ± 5.87 | 79.61 ± 1.58 | 76.49 ± 2.87 |     |
> > |  GCNII   | 77.86 ± 3.79 | 80.39 ± 3.40 | 77.57 ± 3.83 |     |
> > |  MixHop  | 73.51 ± 6.34 | 75.88 ± 4.90 | 77.84 ± 7.73 |     |
> > |  WRGAT   | 81.62 ± 3.90 | 86.98 ± 3.78 | 83.62 ± 5.50 |     |
> > | GraphCON | 75.14 ± 4.95 | 84.90 ± 2.64 | 80.00 ± 3.66 |     |
> > |  HamGNN  | 76.49 ± 5.10 | 83.92 ± 4.87 | 81.62 ± 6.22 |     |
> >
> > __Table R1-Corrected:__ Node classification accuracy(%) on heterophilic datasets under 10 fixed
> > 48%/32%/20% splits taken from [2]. The results are reported from [7] except GraphCON and HamGNN which have been run by us.
> >
> > [2] Pei, Hongbin, et al. "Geom-gcn: Geometric graph convolutional networks." arXiv preprint arXiv:2002.05287 (2020). https://arxiv.org/pdf/2002.05287.pdf
> >
> > [7] Luan, Sitao, et al. "Revisiting Heterophily For Graph Neural Networks." arXiv preprint arXiv:2210.07606 (2022). https://arxiv.org/pdf/2210.07606.pdf

---

> ### Author Response · Authors · 2022-11-23
> **Discussion with Reviewer FBem**
>
> Dear Reviewer FBem, we really thank you for pointing out several weaknesses of our paper e.g. the motivation, the paper organization, and experimental settings. We have made changes from your valuable comments and believe our paper has been improved by using your feedback. We are looking forward to discussing any further questions or suggestions!
>
> Thank you sincerely for your time and effort in reviewing our paper!

---

### Official Review · Reviewer_X8ht · 2022-10-24

**Confidence:** 2
**Correctness:** 3
**Technical Novelty And Significance:** 2
**Empirical Novelty And Significance:** 2
**Recommendation:** 6

**Clarity, Quality, Novelty And Reproducibility:**

 + In terms of clarity, overall, the paper is well written; with the right background knowledge (about Hamilton flows and differential geometry), the paper is not too hard to follow.

 - In terms of novelty, most of the novelty comes for the paper  [Chen et al, 2021] which first proposes directly learning the sympletic 2-forms.  The paper  [Chen et al, 2021]  focuses primarily learning unknown Hamiltonian equations from sampled datasets.  This paper applies to "general" graph learning problems, and investigates several symplectic two-forms.

 +/-  In terms of quality, the paper could have done a better job in justifying why the Hamiltonian flow framework is the right one for many graph learning problems. Given that the Hamilton flows come from physics with certain "conservation laws", what do "Hamiltonian flows" and "sympletic" 2-forms for non-physics settings, e.g., "social network analysis" problems using citesser and Cora datasets, capture? Otherwise, it still seems to be a "black" magic.  Although the authors claim that the proposed HmmGNN helps advoid oversmoothing, there are no theoretical justifications for it.

Incidentally, while the authors in  [Chen et al, 2021]  claim that their method is a "coordinate-free" framework from learning symplectic forms, it is in fact not --- it merely does not assume the specific Darboux coordinate system, and instead assumes the 2-form is expressed in a general local coordinate system. Any time one writes any form in terms of partial differentials, one needs to use a local chart  -- thus one can computations (locally). For the examples in the paper, it seems that the authors assume that the Hamiltonian equations can be written in a single local chart (i.e., everything operates in a Eucliean space), The paper does not address the general manifold where one needs multiple (local) charts, thus chart transitions are needed to ensure consistency of the learned 2-forms.

  +/- The authors provide code and other material as "supplementary material." While the authors show results that illustrate better results, I am not sure these are necessarily fair comparisons. I would have liked to see addition information regarding the number of parameters used, training time, etc.



**Details Of Ethics Concerns:**

There are no ethics concerns.

**Strength And Weaknesses:**

Strengths

   + While based on the "Neural symplectic form" first developed in  [Chen et al, 2021], the paper appears to generalize to graph learning problems.
   + The paper provides right amount of background and mathematical preliminaries to make the paper accessible.
   + The terms and notations are clearly defined.


Weaknesses:
   - The model architecture in section 4.3 could be better explained. Reading the paper and supplementary material only do not provide the reviewer a good idea how the proposed model works. One has to go back to  [Chen et al, 2021] to learn how simpletic forms are actually learned.
   -  While the evaluation results show the proposed HmmGNN outperforms some of the state-of-the-art, I would have liked to better explain why Hamilton flows make sense for the node classifications using Citeseer and Cora datasets. For example, what does  Poincaré 2-form intuitively capture here?

**Summary Of The Paper:**

Extending recent works on Hamiltonian Neural Networks (and Riemannian Manifold GNNs), the paper considers the graph as a discretization of an underlying manifold, and construct a Hamiltonian neural network architecture (HamGNN) for graph learning problems. Building on work  in [Chen et al, 2021],  The Hamiltonian neural network architecture learns the symplectic form defining  the Hamilton flow associated with the Hamiltonian scalar function on its cotangent bundle. The authors investigate several symplectic forms and empirically demonstrate that the proposed approach achieves better node classification accuracy than popular state-of-the-art GNNs.

**Summary Of The Review:**

Building on work  in [Chen et al, 2021], the paper considers the graph as a discretization of an underlying manifold, and construct a Hamiltonian neural network architecture (HamGNN) for graph learning problems.  The authors investigate several symplectic forms and empirically demonstrate that the proposed approach achieves better node classification accuracy than popular state-of-the-art GNNs. The authors could have done a better job in justifying why the proposed framework (that originates from solving physics problems) is a general fit for other non-physics graph learning problems.

---

> ### Author Response · Authors · 2022-11-19
> **Responses to Clarify Reviewer 1 X8ht's Weakness**
>
> $\textbf{Weakness 1: The model architecture in section 4.3 could be better explained. Reading the paper and supplementary}$
> $\textbf{material only do not provide the reviewer a good idea how the proposed model works. One has to go back}$
> $\textbf{to [Chen et al, 2021] to learn how simpletic forms are actually learned.}$
>
> **Response to Weakness 1:** We have rewritten the manuscript in which we have emphasized the motivations in a more intuitive manner at the beginning, significantly condensed the introduction of differential geometry, and provided more details about the model architecture.
>
> $\textbf{Weakness 2: While the evaluation results show the proposed HmmGNN outperforms some of the state-of-the-art,}$
> $\textbf{I would have liked to better explain why Hamilton flows make sense for the node classifications using Citeseer and}$
> $\textbf{Cora datasets. For example, what does Poincaré 2-form intuitively capture here?}$
>
> **Response to Weakness 2:** Thank you for your insight here. In the revision, we have added motivation I and motivation II to better explain how the HamGNN relates to hyperbolic GNNs. First, we claim that (cf. Motivation I) hyperbolic exponential map is a solution under fixed metric to the Riemannian geodesic equation and it enjoys a simple closed-form and thus has been widely applied for the embedding task. The geodesic equation is given in equation (8). Furthermore, the geodesic equation can be reformulated as a Hamiltonian equation (cf. equation (13) in Motivation II). In other words, the geodesic equation can be considered as a special case of the Hamiltonian equation. Based on this understanding, we can directly learn the graph node embedding using Hamiltonian equations where we treat the pseudo-Riemannian metric as a learnable diagonal matrix (cf. equation (20) in Section 4.2.1) rather than a fixed metric as assumed in hyperbolic GNNs. As a result of this adaptive metric, HamGNN is capable of learning the potential embedding space of different structured datasets.

---

> ### Author Response · Authors · 2022-11-19
> **Responses to Clarify Reviewer 1 X8ht's "Clarity, Quality, Novelty And Reproducibility"**
>
> $\textbf{Q1: In terms of quality, the paper could have done a better job in justifying why the Hamiltonian flow framework is the right one for}$
> $\textbf{many graph learning problems. Given that the Hamilton flows come from physics with certain "conservation laws", what do}$ $\textbf{"Hamiltonian flows" and "sympletic" 2-forms for non-physics settings, e.g., "social network analysis" problems using citesser}$ $\textbf{and Cora datasets, capture? Otherwise, it still seems to be a "black" magic. Although the authors claim that the proposed }$
> $\textbf{HmmGNN helps advoid oversmoothing, there are no theoretical justifications for it.}$
>
> **Response 1:**
> Thank you for your criticism. Please refer to the two main motivations in our revision for why Hamiltonian flow is the right one for graph learning problems. Since this paper is focusing on learning a flexible node embedding, we motivate HamGNN from the aspect of geometry instead of its physics features.
> However, the conservation law could be one reason why HamGNN can mitigate the oversmoothing problem, and some existing analysis can be found in [R1]. Since the oversmoothing problem is not this paper's main focus, we do not expand on this point and will leave the theoretical analysis for future work.
>
> [R1] T. Konstantin Rusch, Benjamin Paul Chamberlain, James Rowbottom, Siddhartha Mishra, and Michael M. Bronstein. Graph-coupled oscillator networks. In Proc. Int. Conf. Mach. Learn., 2022.
>
> $\textbf{Q2: Incidentally, while the authors in [Chen et al, 2021] claim that their method is a "coordinate-free" framework from}$ $\textbf{learning symplectic forms, it is in fact not --- it merely does not assume the specific Darboux coordinate system, and instead}$ $\textbf{assumes the 2-form is expressed in a general local coordinate system. Any time one writes any form in terms of partial }$
> $\textbf{differentials, one needs to use a local chart -- thus one can computations (locally). For the examples in the paper, it seems that the}$
> $\textbf{authors assume that the Hamiltonian equations can be written in a single local chart (i.e., everything operates in a Eucliean space), }$
> $\textbf{The paper does not address the general manifold where one needs multiple (local) charts, thus chart transitions are needed to}$ $\textbf{ensure consistency of the learned 2-forms.}$
>
> **Response 2:** You are right. We do assume a single local chart on which all the operations are performed. However, it is really not a trivial problem to consider multiple local charts and establish a connection between them using different smooth transition maps. To do this, one needs to restrict any two neighboring charts to the intersection of their domains. We think it deserves another paper to fully address it.
> Finally, we would like to point out that all the literature for the graph node embedding task has this implicit single but large enough chart assumption. In the revision, we have stated this assumption clearly.
>
> Regarding the "coordinate-free" concerns, we agree with the reviewer's point. In our opinion, the reference [Chen et al, 2021] essentially is targeted at learning a coordinate transformation that is equivalent to learning the symplectic 2-form.
> In our revision, we now mainly focus on the Hamiltonian equation (15)
> that is expressed under the specific Darboux coordinate system. In section 4.2.5 in the revision, we argue whether the learnable $H_{\mathrm{net}}$ is able to learn the energy representation under the chosen chart coordinate system and whether the learnable symplectic 2-form is necessary. The answer is provided in section 5.1 when "comparing the HamGNN using (20) and (23)". We find that the learnable symplectic 2-form indeed does not provide us obvious benefits. We highly recommend the reviewer to section 4.2.5 and section 5.1 for our discussions.
>
> $\textbf{Q3: The authors provide code and other material as "supplementary material." While the authors show results that illustrate better}$
> $\textbf{results, I am not sure these are necessarily fair comparisons. I would have liked to see addition information regarding the number}$ $\textbf{of parameters used, training time, etc.}$
>
>
> **Response 3:**
> Thank you for your feedback. We have provided more detailed instructions for the reproduction of our experiments in the paper appendices with all neural layers in Table 7 in the paper. Furthermore, since we have thoroughly modified the "Proposed Framework" section with a sufficient discussion about the motivation, model architecture, and experimental setups, we hope the reproducibility is now been improved. In the attached Supplementary Material, we also provide sufficient instructions with newly added link prediction experiments and the code for new graph datasets. We hope the details provided in the revision can address the reviewer’s concerns.

---

> ### Author Response · Authors · 2022-11-19
> **Responses to Clarify Reviewer 1 X8ht's "Summary Of The Review"**
>
> **Response:** Thank you very much for your insightful comments! Based on your comments, we have made significant changes in our revision. We really hope our rebuttal responses, together with the paper revision, could make our work more accessible for all readers, including the reviewers!

---

> ### Author Response · Authors · 2022-11-23
> **To Reviewer X8ht**
>
> Dear Reviewer X8ht, we would like to express our thanks for your helpful comments which have definitely made our paper better! Could we kindly ask whether our point-by-point responses and the uploaded paper revision have addressed your main concerns about our work? Thank you again for your efforts in raising valuable questions and improving our paper!

---

> ### Author Response · Authors · 2022-11-29
> **Thanks to Reviewer X8ht**
>
> Dear Reviewer __X8ht__,
>
> Thanks for your support! We deeply appreciate your comments that have been used to improve our paper revision. We would like to know whether our response addresses your concerns. We are also happy to answer if you have any further questions.
>
> Thank you!
>
> Best,
>
> Paper5782 Authors

---

> > ### Author Response · Authors · 2022-11-30
> > **Discussion with Reviewer X8ht**
> >
> > Dear Reviewer __X8ht__,
> >
> > We deeply appreciate your comments that have been used to improve our work. We would like to know whether our responses have addressed your new concerns. We are also happy to answer if you have any further questions.
> >
> > Thank you!
> >
> > Best,
> >
> > Paper5782 Authors

---

### Author Response · Authors · 2022-11-19
**Summary of Revision**

We sincerely thank all the reviewers for their helpful comments and
feedback! We have now uploaded the rebuttal version of our paper where
major changes are highlighted in blue.

Here is the summary of the major changes we made in the revision:
>__1. We modify the abstract which may cause confusion about the mention in
    passing of oversmoothing and oversquashing. We emphasize in the
    revision about our main objective and achievements to make our
    contribution and topic clearer.__

>__2.  We restate our contributions in the "Introduction" section.__

>__3.  We have removed the redundant content in the "Preliminaries" section from the paper. Instead, we save our words from differential geometry concepts and take a new succinct perspective to introduce
    Hamiltonian mechanics. We include much more motivation and explanation of our work in the new "Motivations and Preliminaries" section.__

>__4.  We thoroughly rewrite the "Proposed Framework" to give more details
    of the model architecture, implementation, and network setting.
    Since the "Preliminaries" has been revised, correspondingly we also
    switch to new descriptions of the concepts used in the framework. We
    continue to provide more discussion for the node embedding task and
    the motivation of our strategy.__

>__5.  For the "Experiments" section, in the revision we now include the
    two standard graph node embedding downstream tasks, the node
    classification task and the newly added link prediction task, to
    better demonstrate our model efficacy.__

>__6.  We have included more graph datasets to better demonstrate that our
    proposed node embedding strategy can automatically learn, without
    extensive tuning, the underlying geometry of datasets with diverse
    geometry. Due to space limitations, we have presented the results in
    the appendices.__

>__7.  We have provided more detailed instructions for the reproduction of
    our experiments in the paper appendices in the revision. In the
    attached “Supplementary Material”, we provide sufficient
    instructions with new link prediction experiments and the code for
    new graph datasets.__

Finally, we would like to sincerely thank all the reviewers again for
their enormous efforts putting in reviewing our paper and raising
valuable questions! Without their valuable comments, we could not
improve our manuscript to its current status.

---

### Author Response · Authors · 2022-11-23
**To all reviewers**

Dear reviewers,

To facilitate exposition, we have modified the rebuttal to include the original comments from all reviewers. We hope this can help when you read our responses.

Thank you so much for your time and effort in reviewing our paper! We hope that our responses can now convince you that our paper is fundamentally improved.

Best,

Authors

---

### Author Response · Authors · 2022-11-29
**gentle reminder for active discussions**

Dear Reviewers and ACs,

We put a great effort into conducting new experiments, writing point-by-point responses to each reviewer's comments, and updating the paper manuscript. We however have only seen feedback from __Reviewer FBem__. Many thanks for his new comments!

We totally understand that each reviewer may be busy with multiple reviewing tasks. However, we have seen that many other ICLR papers are involved in active discussions between the reviewers, area chairs, and authors. We indeed think such active discussions help the reviewing process and will accelerate the fair understanding of the work. We welcome any new questions that need to be answered, and we really hope reviewers (or even ACs) can give some comments for our rebuttal.

Thank you!

Best,

Paper5782 Authors

---

### Decision · Program_Chairs · 2023-01-20

**Decision:**

Reject

**Justification For Why Not Higher Score:**

lack of SOTA results and clear advantage over related DE-based methods

**Justification For Why Not Lower Score:**

N/A

**Metareview: Summary, Strengths And Weaknesses:**

The paper proposes a GNN architecture based on Hamiltonian equations. It falls under the broad recent category of physics-inspired/DE-based GNN models.

The reviewers complained about insufficient clarity and poor motivation of the use of a differential-geometric mathematical model (not very familiar to the broad ML audience). This is exacerbated by lack of SOTA results (other methods from a same class often perform better). The rebuttal addressed some questions but overall we believe the paper is below the ICLR bar.